# History, Rats, Fleas, and Opossums. II. The Decline and Resurgence of Flea-Borne Typhus in the United States, 1945–2019

**DOI:** 10.3390/tropicalmed6010002

**Published:** 2020-12-28

**Authors:** Gregory M. Anstead

**Affiliations:** Medical Service, South Texas Veterans Health Care System and Department of Medicine, University of Texas Health San Antonio, San Antonio, TX 78229, USA; anstead@uthscsa.edu

**Keywords:** rats, flea, opossums, *Rickettsa typhi*, *Rickettsia felis*, insecticide, rodenticide

## Abstract

Flea-borne typhus, due to *Rickettsia typhi* and *R. felis*, is an infection causing fever, headache, rash, and diverse organ manifestations that can result in critical illness or death. This is the second part of a two-part series describing the rise, decline, and resurgence of flea-borne typhus (FBT) in the United States over the last century. These studies illustrate the influence of historical events, social conditions, technology, and public health interventions on the prevalence of a vector-borne disease. Flea-borne typhus was an emerging disease, primarily in the Southern USA and California, from 1910 to 1945. The primary reservoirs in this period were the rats *Rattus norvegicus* and *Ra. rattus* and the main vector was the Oriental rat flea (*Xenopsylla cheopis*). The period 1930 to 1945 saw a dramatic rise in the number of reported cases. This was due to conditions favorable to the proliferation of rodents and their fleas during the Depression and World War II years, including: dilapidated, overcrowded housing; poor environmental sanitation; and the difficulty of importing insecticides and rodenticides during wartime. About 42,000 cases were reported between 1931–1946, and the actual number of cases may have been three-fold higher. The number of annual cases of FBT peaked in 1944 at 5401 cases. American involvement in World War II, in the short term, further perpetuated the epidemic of FBT by the increased production of food crops in the American South and by promoting crowded and unsanitary conditions in the Southern cities. However, ultimately, World War II proved to be a powerful catalyst in the control of FBT by improving standards of living and providing the tools for typhus control, such as synthetic insecticides and novel rodenticides. A vigorous program for the control of FBT was conducted by the US Public Health Service from 1945 to 1952, using insecticides, rodenticides, and environmental sanitation and remediation. Government programs and relative economic prosperity in the South also resulted in slum clearance and improved housing, which reduced rodent harborage. By 1956, the number of cases of FBT in the United States had dropped dramatically to only 98. Federally funded projects for rat control continued until the mid-1980s. Effective antibiotics for FBT, such as the tetracyclines, came into clinical practice in the late 1940s. The first diagnostic test for FBT, the Weil-Felix test, was found to have inadequate sensitivity and specificity and was replaced by complement fixation in the 1940s and the indirect fluorescent antibody test in the 1980s. A second organism causing FBT, *R. felis*, was discovered in 1990. Flea-borne typhus persists in the United States, primarily in South and Central Texas, the Los Angeles area, and Hawaii. In the former two areas, the opossum (*Didelphis virginiana*) and cats have replaced rats as the primary reservoirs, with the cat flea (*Ctenocephalides felis*) now as the most important vector. In Hawaii, 73% of cases occur in Maui County because it has lower rainfall than other areas. Despite great successes against FBT in the post-World War II era, it has proved difficult to eliminate because it is now associated with our companion animals, stray pets, opossums, and the cat flea, an abundant and non-selective vector. In the new millennium, cases of FBT are increasing in Texas and California. In 2018–2019, Los Angeles County experienced a resurgence of FBT, with rats as the reservoir.

## 1. Introduction

Flea-borne typhus (FBT), also known as murine typhus, is an infection caused by *Rickettsia typhi* and *R. felis*. The infection causes fever, headache, rash, and diverse organ manifestations. While most cases are self-limited, 26–28% have complications and up to one-third require intensive care [1]. Flea-borne typhus is transmitted to humans by the inoculation of a bite site, a skin abrasion, or mucous membranes with feces from fleas infected with these rickettsiae or by a flea-bite [2,3]. This is the second part of a two-part series describing the historical epidemiology of FBT in the USA. Part I included: a clinical overview; an account of the discovery and relationships of the vectors and reservoirs; and an examination of the factors that led to an increase in the number of FBT cases from 1920 to 1944. In the 1920s and 1930s, the black and brown rats (*Rattus rattus* and *Ra. norvegicus*, respectively) were found to be reservoirs of FBT and various species of fleas (Oriental rat flea (*Xenopsylla cheopis*), chicken flea (*Echidnophaga gallinacea*), European mouse flea (*Leptopsylla segnis*), cat flea (*Ctenocephalides felis*), and Northern rat flea (*Nosopsyllus fasciatus*)) were determined to be the vectors. The disease was recognized to be a threat to health in the southern United States in the 1920s and an increasing number of cases were observed in the 1930s and early 1940s. About 42,000 FBT cases were reported in the USA between 1931–1946 and the infection peaked in 1944 at 5401 cases. The dramatic rise in the number of cases from 1930 through 1944 was due to the diversification of Southern agriculture away from cotton to crops palatable to rodents, the displacement of the smaller black rat by the larger brown rat in many areas, the poor condition of housing during the Great Depression and World War II, and shortages of effective rodenticides and insecticides during World War II. Attempts to control FBT in the 1930s and early 1940s by rat proofing, fumigation, and rat extermination were unsuccessful [4]. Part II will discuss: the innovations in insecticide and rodenticide technology occurring during and after World War II, the public health programs instituted for typhus and rodent control in the post-war period, the improvement in social and economic conditions in the endemic areas after World War II, advances in the diagnosis and treatment of FBT, the change in the epidemiology of FBT from a rodent flea-borne disease to a cat flea-borne disease, the recognition of the opossum (*Didelphis virginiana*), cats, and dogs as a significant reservoirs, the discovery of a second causative bacterium (*R. felis*), and epidemiologic trends in the persistent endemic areas of the USA (Texas, California, and Hawaii).

## 2. The Decline of Flea-Borne Typhus, 1945–1990

After World War II, there was a precipitous decline in the number of cases of FBT in the USA. From its peak of 5401 cases in 1944 [5,6], the number of FBT cases declined to only 98 in 1956. The number of reported cases of FBT reached a nadir of 18 in 1972 (Table 1) [2,7,8,9]. The factors responsible for this rapid decline in the number of cases of FBT deserve analysis. However, a caveat is that it is difficult to directly compare the number of cases of FBT over decades due to differences in case ascertainment and reporting and serologic techniques. Nevertheless, general epidemiologic trends may be discerned.

In October 1942, the US Dept of Agriculture (USDA) received a sample of dichlorodiphenyltrichloroethane (DDT) from Geigy Co. in Switzerland. Among the hundreds of chemicals that the USDA tested as insecticides during World War II, DDT stood out for its potency, broad-spectrum insecticidal activity, residual action (3–4 months), and its ability to kill insects on contact [10,11]. DDT dust (5–10% DDT mixed with an inert carrier, such as talc, kaolin, or pyrophyllite) could be conveniently applied either manually or by machine to rat runs [12]. Thus, DDT had distinct advantages compared to previous insecticides, such as rotenone, pyrethrum, nicotine, and the thiocyanates.

DDT immediately showed its value against insect-borne disease on massive scale. In the summer of 1943, Naples, Italy, lay in ruins due to German sabotage and torrents of Allied bombs. Much of the populace resorted to living in abandoned tufa quarries and air-raid shelters. Under these overcrowded and squalid conditions, an outbreak of louse-borne (epidemic) typhus (due to *R. prowasekii*) ignited in Naples in November of 1943. In December 1943, the Allies instituted a vigorous campaign to dust the entire populace of Naples with DDT and other lousicides, and a potentially catastrophic outbreak of epidemic typhus was averted [13]. DDT was also used to combat an epidemic of plague in French West Africa in April 1944 [10]. With these successes, DDT thereby ushered in a new era of ectoparasite control [11] and it was soon unleashed against FBT on the home front. San Antonio, Texas, was selected as an ideal demonstration site for this new weapon against fleas because cases of FBT had nearly tripled there during World War II, with 32 cases in 1943 and 91 cases in 1944 [14]. With its four military bases, San Antonio experienced dramatic growth during the war, with its population expanding from 253,854 in 1940 to 335,000 in 1945 [15,16]. The corrupt San Antonio city government of the time was incapable of delivering adequate sanitary services to accommodate this influx of people [17]. To implement the project, a Typhus Advisory Committee was established, chaired by a renowned tropical medicine specialist, Col. Charles F. Craig, formerly of the US Army Medical Corps [14,18,19]. Studies of the epidemiology of FBT in San Antonio and the impact of DDT dusting were conducted by David E. Davis of the United States Public Health Service (USPHS). Davis had just completed a two-year stint investigating yellow fever vectors in Brazil for the Rockefeller Foundation [20]. Davis was assisted in the San Antonio endeavor by Morris Pollard (U.S. Army Veterinary Corps), Dr. Lewis Robbins (San Antonio City Health Dept), and Dr. Elsmere Rickard (Rockefeller Foundation), who investigated the human cases [18].

Davis and Pollard initially performed a serosurvey with the complement fixation test on 4219 food handlers in San Antonio and found 3.5% positivity [21]. In a preliminary study of rodent infestation, Davis and Pollard divided the city into six zones. They found about equal numbers of *Ra. rattus* and *Ra. norvegicus*. They also observed that the most crowded zone, inhabited by the poorest residents, had the greatest abundance of rats and the highest percentage of FBT-seropositive rats (52% and 66% for *Ra. rattus* and *Ra. norvegicus*, respectively). In the most affluent zone, rats were rare, and the rates of FBT seropositivity were much lower (13% and 16% for *Ra. rattus* and *Ra. norvegicus*, respectively). Human cases of FBT in San Antonio were more common in areas with abundant rats [18]. Davis then assessed the average rodent flea burden (total no. of fleas from the rats examined/total no. of rats examined). Rats were collected from the one-story frame houses and corner groceries on the impoverished south side of the city; *X. cheopis* and *L. segnis* were the most common fleas. *Xenopsylla* populations peaked in May through August, when 86% of rats harbored these fleas, and the average flea burdens were 12.6 and 6.3 for *Ra. norvegicus* and *Ra. rattus*, respectively. For *L. segnis*, the maximum abundances occurred in March and April, and it was virtually absent in July and August. *Ctenocephalides felis* fleas were most common from May to June. The maximum abundance of fleas occurred in May and June, but because of different peaks for *X. cheopis* and *L. segnis*, fleas were common in San Antonio for six months of the year. Seropositive rats carried more fleas than seronegative ones. Large numbers of fleas were found on some rats, with one hosting 423 fleas [22]!

Before DDT dusting in San Antonio, the average FBT seropositivity rates of *Ra. rattus* and *Ra. norvegicus* were 30% and 70%, respectively, and the average flea burdens were 2.6 and 10.3, respectively [14]. From April through August of 1945, over 22,000 premises were dusted with 2700 kg of 10% DDT in pyrophyllite. After dusting, seropositivity rates of *Ra. rattus* and *Ra. norvegicus* dropped to 13% and 27%, respectively, and the average flea burdens decreased to 1.9 and 5.8, respectively [14]. There were 23 cases of FBT in the untreated areas of San Antonio but only four cases in treated areas, and two of these cases manifested in houses that had been missed by the dusting crews. Davis concluded that DDT dusting was effective, but it represented an auxiliary method of typhus control when an area cannot be economically rat proofed or when control must be achieved rapidly [14].

Due to the surge in Texas FBT cases, from 13 in 1930 to 1740 in 1944, State Health Officer George Cox and U.S. Rep. Albert Thomas of Houston appealed to Maj. John Essex of the USPHS for assistance and in July of 1945 the state was awarded a $500,000 grant and 51,000 kg of DDT to wage war on typhus. The components of the program were: application of DDT to rat-infested premises and rodent abatement, through education, extermination, proper refuse disposal, and rat-proofing. Thirty-six Texas counties and eight cities (Austin, Corpus Christi, Dallas, Fort Worth, Houston, Laredo, Lubbock, and San Antonio) were approved for the program [23]. In 1947, Dr. Cox appealed to every Texan to cooperate with strict rodent control measures [24]. 

On 1 July 1945, the cooperative state-federal typhus control program was initiated by the USPHS and was fully operational by March 1946. It was not feasible to treat all counties with FBT cases, so counties with 50 or more cases during 1940–1944 or ten or more cases in 1944 received the highest priority. The dusting of urban businesses was emphasized, but residential and rural premises in highly endemic areas were also included [25]. During 1945 and 1946, premises were dusted two to four times in most locales, with the number of dustings in subsequent years reduced to one or one in alternate years, or less, as time went on. As flea and rodent control was established, instead of wide DDT distribution, pin-point dusting was done in areas with persistent cases, large Oriental rat flea populations, or heavy rat infestations [26]. In May of 1946, the USPHS released a 28-page pamphlet *DDT for the Control of Murine Typhus Fever* in which the properties of DDT, application methods, assessment of results, and its integration with rodent control were described (Figure 1) [27]. 

Programs were implemented in 122 of the highest FBT reporting counties in nine southeastern states (Alabama, Florida, Georgia, Louisiana, Mississippi, North Carolina, South Carolina, Tennessee, and Texas) in 1946 and the first half of 1947 (Figure 2 and Figure 3). In 1944, these 122 counties reported 3767 cases, accounting for 71% of all American FBT cases. In 1946, the number of cases had decreased by 51% in 1838. In the ten highest reporting counties, the number of cases fell from 1074 in 1944 to 395 in 1946. In 460 counties not dusted with DDT, FBT cases dropped 7% in 1946, but rebounded 10% in 1947 (see Figure 3 for a comparison of dusted and non-dusted counties). The diminution in FBT cases correlated with a decline in flea populations; based on counts from 17,000 rats, the number of rat fleas dropped 84% in the dusted areas [28]. When the number of fleas per rat was below three, there was little spread of FBT to the human population [29].

In September 1945, the USPHS initiated FBT surveillance in the rat population of Galveston, TX; only *Ra. norvegicus* was present. Initially, 63% of Galveston rats were seropositive for FBT. After a six-month dusting program in 1946, the seropositivity rate dropped to 32.7%. By 1947, the percentage of rats infested with *X. cheopis* fell from 67 to 15, and the fleas per rat decreased from 7.4 to 1.1. In this study, the application of DDT had no effect on populations of possible intermurid vectors, the tropical rat mite *Ornithonyssus bacoti* and the spiny rat louse *Polypax spinulosa* [30]. Likewise, in rural Georgia, DDT dusting was effective against *X. cheopis* and *L. segnis*, but ineffective against *P. spinulosa* and *O. bacoti* [31].

The USPHS Thomasville (Georgia) Typhus Investigation Project was conducted April 1946 through September 1947 to determine the effectiveness of DDT dusting as an FBT control measure; three counties in southwest GA were selected: Brooks, Thomas, and Grady. Rat runs in the former two counties were dusted with 10% DDT in pyrophyllite, whereas Grady Co. remained untreated [32]. In Thomas and Brooks counties, after DDT dusting the percentage of seropositive rats dropped from 51 to 6.5 and the percentage of rats harboring *X. cheopis* and *L. segnis* declined from 35 and 56 to 15.4 and 3.9, respectively. Cases of FBT in the two dusted counties plummeted from 100 in 1945 to only three in 1949. Even two years after DDT dusting ceased, suppression of FBT persisted. By comparison, in Grady Co., there were 46 cases of FBT in 1945 and 25 cases in 1949 [33].

Grady Co. was later used as a test site to determine how quickly rat and flea re-infestation occurred in a rural area after rodenticide/insecticide application [32,34]. Between July 1953 and May 1954, rat poisoning and DDT dusting was performed in one area of the county. Surveillance was then conducted over the next three years. Three months after rodent and flea eradication measures were completed, about 5% of the previously cleared farms were re-infested with rats. At subsequent annual inspections from 1955 to 1957, infestation rates grew to 29%, 33%, and 42%, respectively, even though some farmers had continuing rodent control activities. Nevertheless, *X. cheopis* flea indices (fleas/rat) remained low (<1) throughout the three-year period. Rates of typhus seropositivity in rats also remained low (4.3, 0.7, 2.9, and 0%, in 1954, 1955, 1956, and 1957, respectively). In contrast, *E. gallinacea* infestation increased, with flea indices of 3.1, 6.0, 7.4, and 3.9 during each successive year. Thus, DDT dusting was much more effective for the control of the Oriental rat flea as compared to the chicken flea. No known cases of FBT occurred during the three-year surveillance period. The study demonstrated that significant rat re-infestation occurred quickly in the absence of continued intensive control measures. Likewise, Davis and Christian reported that once rodent control ceases in a particular area, the high fecundity of rats cause the population to rebound [35]. Furthermore, when the dominant vector, the Oriental rat flea, is present in low numbers, other potential vectors are not sufficient to maintain high rates of typhus within the rat population [32,34].

In 1952, there were reports of DDT resistance in the cat flea in the USA, but the Oriental rat flea showed no significant resistance through at least 1956. Despite the emergence of insect resistance within a decade of its introduction, the advent of DDT changed the entire paradigm of pest control. Organic chemists, inspired by the success of DDT, created a diverse array of synthetic pesticides. These compounds were of higher potency, versatility, and consistent quality compared to the older botanical and inorganic insecticides, and could be produced cheaply on a massive scale. By the early 1950s, several new chlor-inated hydrocarbons (chlordane, lindane, and heptachlor), as well as the organophosphate malathion were now available to circumvent insecticide resistance [36]. After World War II, there were thousands of trained pilots and surplus trainer airplanes available. Aerial dusting thus became a widely used method to slather the new insecticides on vast tracts of cropland [37]. The ensuing decades would lead to the expansion of the available organophosphates and new classes of insecticides, such as the carbamates and synthetic pyrethrins [38]. While DDT was banned in the USA in 1972, by 1976, there were 200 different insecticides on the American market [39]. In the 1970s, the American South accounted for two-thirds of insecticide usage in the USA [40]. From 1950 to 2002, pesticide usage grew 50-fold in the USA and 1.1 million metric tons of pesticides were used each year [41]. The widespread use of insecticides in agriculture, markets, households, and grain storage facilities reduced flea populations and decreased the risk of FBT [5]. In 1948, the Florida Board of Health reported that 80% of householders were using DDT, and they concluded that this contributed to the reduction of FBT in the state [42].

Even in areas that did not receive dusting, cases of FBT declined after 1945. Joseph Smadel proposed that the widespread agricultural and domestic use of insecticides after World War II also served to interrupt the chain of transmission of *R. typhi*. A similar phenomenon (adventitious control) occurs with mosquito-borne infections, i.e., agricultural insecticides use leads to decrease the number of mosquitoes, which in turn leads to a decreased number of mosquito-borne infections [5]. 

## 3. The Advent of New Rodenticides and Integrated Rodent Control 

World War II also stimulated the development of new rodenticides. By the 1920s, rats were known to be reservoirs of multiple infections (plague, leptospirosis, trichinosis, and rat bite fever), in addition to causing the destruction of stored food [43]. When the Allies entered World War II, it became imperative to protect food resources and war workers in crowded cities from the ravages of the rat. While rodenticides had been used for hundreds of years, the modern science of effective rodenticide use commenced in England in 1939, when Dennis H. Chitty and Charles S. Elton of the Bureau of Animal Population at Oxford University began studying the feeding behavior of rats on poison baits [44,45]. One advance in rat control arising from this work was the principle of pre-baiting, i.e., introducing poison-free bait stations into the area so that the rats became accustomed to them before presenting poisoned bait [46]. 

Meanwhile, in the early 1940s, Baltimore (Maryland) was overrun by rats. Pre-war Baltimore had the third worse housing of any American city and an influx of war workers further exacerbated housing and sanitation problems. However, an unlikely figure emerged from behind the walls of the ivory tower to confront the rats born of Baltimore’s urban squalor. While studying the genetics of the human ability to taste phenylthiourea (PTU), biologist Curt Richter of Johns Hopkins University observed that PTU was toxic to rats, but not readily ingested by them. This inspired the idea of creating a new rodenticide based on PTU. The Dupont Company then prepared 200 analogs of PTU, and the most suitable of these compounds as a rodenticide was alpha-naphthylthiourea (ANTU). ANTU was found to be 100-fold more toxic than red squill for brown rats and to have twice the toxicity of thallium sulfate. In 1942, Richter, armed with ANTU and funding from the city of Baltimore, initially unleashed a 200-city block deratization campaign. Ecologist John T. Emlen Jr moved to Johns Hopkins University during World War II and joined Richter’s team. Emlen focused on understanding rat population dynamics. From May 1943 to August 1945, 6960 blocks were treated with ANTU and Richter estimated that more than a million rats succumbed [44]. A subsequent operation utilizing ANTU, community sanitation, and rat proofing, was conducted in 80% of the residential blocks in Baltimore in 1945–1946. Reductions of rat populations of 85% to 95% were achieved; such decimated populations required 15 to 44 months to return to their original levels. Thus, annual campaigns were sufficient to maintain the suppression of the rat populations [47]. As a result of these successful rat elimination campaigns, ANTU was approved in May 1946 in the USA for rodent control [44].

The development of a second wartime rodenticide was stimulated by the events on the other side of the globe. As Allied troops clawed their way through the tropical vegetation to regain the territory conquered by the Japanese, they encountered an odious creature, the scrub typhus mite, *Leptotrombidium* sp. The bite of one of these mites infected with the scrub typhus organism (*Orientia tsutsugamushi*) produces a febrile illness, in which death may ensue from pneumonitis, myocarditis, or encephalitis. The illness typically lasted 14–28-days and required a prolonged convalescence [48]. Scrub typhus was feared more than malaria because there was no treatment and up to 70% of patients died [49]. By the end of the war, 16,000 Allied troops (and 20,000 Japanese) were stricken with scrub typhus; 639 Allied soldiers died of the infection [48]. 

As rodents hosted the *Leptotrombidium* mites, rodenticide development became a military priority. With the wartime limitations on supplies of many of the usual rodenticides [4], the Wildlife Research Laboratory of the US Fish and Wildlife Service embarked on a crash program for the development of new agents [50]. ANTU was still undergoing trials and was only effective against brown rats, so it was not ideal [51]. Over two years, scientists at the Wildlife Research Laboratory and chemists at the Patuxent Research Refuge tested more than a thousand compounds for toxicity to rodents; in 1945, they discovered that sodium fluoroacetate (Compound 1080) was the deadliest rodenticide ever devised [52]. A standard method to use Compound 1080 was to construct a poison station with an aqueous solution of 1080 in a tip-proof cup. The floor area around the 1080 station was coated with a thin layer of DDT dust to kill any fleas abandoning the poisoned rats (Figure 4) [51,53].

Thus, an array of rodenticides was available in the immediate post-war period, including ANTU, zinc phosphide, arsenic trioxide, red squill, thallium sulfate, and 1080 [51,54]. While these compounds were effective rodenticides, some of these (e.g., zinc phosphide, 1080, and thallium sulfate) were so highly toxic that their use was restricted to trained personnel [51,55]. In terms of dead rodent recovery, Compound 1080 was ten-times more effective than any other rodenticide available in 1945–1946 [56], but it was so toxic that owners of the treated premises had to sign a release form absolving health authorities of responsibility in the event of an accidental poisoning [57]. Compound 1080 was manufactured with the addition of the dye Nigrosine so that an aqueous solution was an unappetizing black color to discourage accidental ingestion. The use of thallium sulfate was limited because of its expense [51]. 

Near the end of World War II, the USPHS shifted personnel that had been engaged in mosquito abatement into typhus control. There were 20 rat-proofing projects underway on 1 July 1945, which quickly expanded to 101 typhus control projects by 15 October 1945, 56 of which used DDT dusting alone and 45 using DDT dusting and rat control. By 1946, the number of USPHS personnel employed in typhus control had grown from 60 to 400 [56]; these workers received the epidemic typhus vaccine, which may have been cross-protective against FBT [58]. Additionally, on 1 July 1946, the Communicable Disease Center (CDC; later to become the Centers for Disease Control) of the USPHS was established [59]. This organization would play a major role in the post-war FBT and rodent control programs.

In 1948, the total rat population of the USA was estimated to be 123 million, compared to 147 million people [60]. However, now equipped with the new post-war rodenticides and exploiting the methodology of pre-baiting, the US Department of the Interior spearheaded a national crusade for domestic rodent extirpation. A National Committee for Rat Control was formed, consisting of executives from the food and restaurant industries. This committee worked in conjunction with the US Fish and Wildlife Service of the Department of the Interior. In March 1948, Interior Secretary Julius Krug offered federal aid to all governors and to the mayors of the 1077 American cities with a population of ≥10,000 for the establishment of rodent abatement programs [61]. Later in 1948, the National Urban Rat Control Campaign was launched in 405 cities [62,63]. The components of the program included: refuse elimination, rat proofing and extermination, the organization of inspections, the enforcement of anti-rat ordinances, and educational and promotional efforts [44,64]. In the mid-1940s, rodent-poisoning campaigns could be rapidly carried out for $100–125 per thousand population or about $11 per building. Rat proofing cost an average of $80 per building but often required two to three years to complete in a community. Additionally, rat proofing was impractical for poorly constructed or maintained buildings [56]. Nevertheless, as a result of the intensive urban rodent control measures, by the late 1940s, FBT had shifted to a predominately rural disease [5,65].

After serving two years with the USPHS on FBT control in San Antonio, David Davis joined the faculty of Johns Hopkins University School of Hygiene and Public Health. Along with biologists John Emlen Jr and John Calhoun, Davis formed the Rodent Ecology Project, which was funded by the Rockefeller Foundation. From 1946–1952, this group studied the ecology, behavior, population dynamics, health, predation, and control of urban rats [66,67,68]. By the early 1950s, there were increasing efforts in American cities to use a multi-modal approach to rodent control with sanitation, elimination of harborage, and rodent-resistant construction as the most important measures; trapping and rodenticide application were secondary methods [44]. There was also increasing integration of the rodent control activities of the public health sector (inspection and enforcement of building, sanitation, and health codes) with environmental remediation and rodenticide application by commercial exterminators [69]. 

In 1949, the CDC released a comprehensive 293-page manual *Rat-Borne Disease: Prevention and Control*. To break the monotony of the scientific presentation, the principles of rodent control were told through the adventures of “Rodney, the rat-ridder” [70]. By 1950, the CDC had distributed 5000 copies of this manual to federal, state, and local health authorities, with plans to distribute an additional 5000 copies [71]. Education for rodent control by public agencies was extended to include the whole family. In 1949, the Florida State Board of Health published a booklet intended for “boys and girls and their parents” entitled *Roddy the Rat: A Story of Typhus Fever and of Ways of Getting Rid of Rats,* which described in graphic novel form how a marauding pack of rats were foiled by individual and community action. The second half of the booklet described practical measures regarding rat trapping and poisoning [72]. By 1950, there were 167 communities in the nine high-incidence FBT states of the American South that had established rat-proofing and anti-rodent sanitation programs. [26].

Throughout the 1940s and into the early 1960s, the CDC (and its predecessor organization, Malaria Control in War Areas) promoted education for FBT control directed at public health officers, producing twelve movies and film strips on rodent control, one film on FBT, one film strip on flea identification, and ten movies and film strips on insecticide usage [73,74,75]. In 1946, the CDC provided a two-week course for personnel engaged in typhus control and a six-week course for project supervisors [76]. In the 1960s, the CDC was offering training courses on: Epidemiology and Control of Vector-Borne Diseases; Insect Control; Advanced Rodent Control; and Insect and Rodent Control [73].

Meanwhile, in the 1920s, cattle in the USA were afflicted by a condition causing fatal hemorrhage. Moldy silage made from sweet clover was implicated, and in 1940, biochemist Karl Link and his graduate students Willard Roberts, Harold Campbell, Charles Huebner, and Mark Stahmann at the University of Wisconsin discovered that the anticoagulant in sweet clover was dicoumarol [77]. Chemical modification of dicoumarol by Link and colleagues led to warfarin in 1948. In a trial against brown rats in Savannah (GA), warfarin proved efficacious, even where 1080 and ANTU had failed [78]. Warfarin was approved as a rodenticide in the USA in 1952 (Figure 5 and Figure 6) [79,80,81,82]. Of all the new rodenticides, warfarin had the most favorable attributes: it was highly effective, inexpensive (a lethal dose costed four cents in the early 1950s), it did not induce bait shyness, and it had a good safety profile [31,51,78,83]. Additionally, because it did not require elaborate safety precautions, it was made available for general consumer use. As an example of its effectiveness, in 1952, the USPHS launched a typhus elimination project in rural Geneva County, Alabama. Prior to implementation, 60% of rural premises were rat-infested; after the application of warfarin, the level of infestation decreased to 14% [84]. Two decades had passed since Dr. John Phair had proclaimed that the control of rural rats was impossible [85]. However, by the late 1950s, using anticoagulant rodenticides, even in rural areas, rats could be controlled effectively and at reasonable cost [31]. By the 1980s, anticoagulants comprised 95% of the rodenticides used in the USA [86].

Forty-eight cities and towns had rodent abatement projects in operation during 1949. One hundred and twenty-one communities had completed their programs and were maintaining rat proofing and environmental sanitation with the assistance of the USPHS and state agencies [26]. The most effective FBT mitigation projects used a combination of DDT, the new rodenticides, and environmental management. In Georgia, after DDT dusting and the application of warfarin to 75,000 premises, the percentage of seropositive rats dropped from 31.6 in 1946 to 6.4 in 1951, and the number of human cases of FBT decreased in parallel, from 1111 in 1945, to 58 in 1951 [87]. The incidence of human FBT in several southwest Georgia counties fell dramatically from 202–218 cases/100,000 residents in 1945 to 5.7–10.5/100,000 in 1949 [88]. To discourage rodent infestation, by the early 1950s, 30 communities in Georgia discontinued open garbage dumping in favor of sanitary landfills [65,87].

In the early 1950s, the seroprevalence of FBT in rats was assessed in several Southern cities. In Corpus Christi (Texas), 304 premises were inspected; only 11 had rats and all were seronegative. In Mobile and Birmingham (Alabama), 436 rats were captured, all seronegative. The sera of rats that had been collected by the rodent control programs throughout the American South from 1946 to the 1952 was tested for FBT reactivity by the CDC. In 1946, 45% of rats were seropositive; by 1952, the rate had dropped to 6.7%. The number of human cases of FBT in the USA dropped to 423 in 1951 (Figure 7) [89]. The USPHS typhus control program was terminated in 1952. It had been a truly monumental endeavor of both public health and science, conducted by thousands of technicians and scientists in nine states over seven years. Millions of rats and fleas had been exterminated, thousands of rats had been phlebotimized for serologic determinations, and ectoparasites by the tens of thousands had been meticulously collected and identified. The end result was a 15-fold decrease in the number of reported FBT cases in the United States [90]. Clearly, the post-war USPHS campaign had achieved its goal of FBT suppression in rodent and human populations.

In 1962, the CDC issued guidelines regarding the sequence of operations to control FBT and plague: (1) rodent surveys in the control area; (2) application of residual insecticides (DDT, dieldrin, or heptachlor); (3) use of anticoagulant rodenticides; (4) rat trapping and poisoning with single-dose high toxicity rodenticides (red squill and zinc phosphide), or burrow gassing; (5) elimination of rodent food sources and harborage; (6) vent stoppage or rat proofing; and (7) continued surveillance and maintenance. The initial application of residual insecticides was intended to achieve a rapid flea kill. The anticoagulant rodenticides were distributed on day 1 because these compounds needed to be consumed for several consecutive days to achieve lethality. Rat-trapping or poisoning with single-dose rodenticides was delayed for several days to allow rodents in the treated areas to pick up the insecticide on their feet or fur and relay it back to their burrows. As the rodents are trapped or succumb to rodenticides, any fleas remaining on the rodents will be killed by the insecticide before they are able to infest other animals, including humans. Environmental sanitation should be improved quickly as flea elimination is achieved. It was recommended that rodent control workers be vaccinated against plague or typhus and wear flea-proof clothing treated with insect repellents. It was advised that vent stoppage work be delayed until the danger of contracting FBT or plague was reduced [91].

In addition to the availability of new insecticides and rodenticides and the implementation of programs for their wide-scale use, social factors also contributed to the decline of FBT. Due to World War II and New Deal policies, economic conditions had considerably improved in the American South by 1945 [93]. The war had transformed the Southern economy by the diversification of agriculture and industry, the migration of excess labor, and urbanization [94]. The total expenditure for Southern war plants was $4.4 billion, which boosted the industrial capacity of the South by 40% [95]. Four billion dollars had been spent on Southern military facilities [94]. By the end of the war, public services and housing began to improve in the urban South [96].

In addition to the industrial development in the American South during the war years, between 1940 and 1944, farm incomes doubled. With the migration of 2.5 million surplus rural laborers north, per capita farm income tripled [94]. The number of the tenant farmers dropped from 1,449,293 before the war to 1,165,279 at the end of the war [95]. The percentage of farms with electricity doubled [97]. While a region in the South would remain economically disadvantaged and pockets of desperate poverty would remain, the post-war years marked the beginning of the transformation of the South from an economic backwater into the more prosperous Sunbelt [98]. 

Housing improvements and urban renewal also contributed to the decline of FBT. After World War II, there was an effort to incorporate rat-proof materials and design features into new buildings [56]. In 1946, Congress enacted the Veterans Emergency Housing Act, which led to the construction of almost one million units of housing. The Housing Act of 1949 provided grants and loans for slum clearance, urban redevelopment, farm housing, and the construction of 810,000 housing units. In the nation as a whole, the quantity of housing classified as dilapidated decreased by half, from 1950 to 1960, by slum remediation, urban renewal, and new construction. Home ownership, associated with better housing conditions, was at an all-time high in 1960 [99]. In the long term, improved sanitation and housing are the most effective means of rodent control [100]. By 1956, as a result of FBT control programs and improvements in housing and economic conditions, only 98 cases were reported in the USA [6]. 

In 1967, President Lyndon Johnson proposed the Rat Extermination and Control Act, but it was defeated in the US House of Representatives. During discussion of the legislation on the floor of the House, the opponents, mostly Republicans and Southern Democrats, ridiculed the legislation, terming it the “civil rats bill.” Amidst laughter, opponents argued that if rat control became a federal concern, next there would be federal aid for snake, squirrel, bug, and blackbird eradication. One opponent, James Haley (Democrat-Florida), suggested “buy a lot of cats and turn them loose.” Strongly endorsing the bill was Martha Griffiths (Democrat-Michigan), who said that rats had killed more persons than “all the generals in history.” Griffiths, whose speech squelched the laughter, said “rats are a living cargo of death and you think it’s funny… If you’re going to spend $75 billion to try to kill off a few Viet Cong, I’d spend $40 million to kill the most devastating enemy that man has ever had” [101]. 

Johnson was undeterred by the defeat of his bill in Congress; he frequently discussed the issue in public and pressed individual legislators for their support. In a 1967 speech Johnson declared “the knowledge that many children in the world’s most affluent nation are attacked, maimed, and even killed by rats should fill every American with shame” [66]. By August 1967, an opinion poll showed that 59% of Americans supported Johnson’s rat control policy [102]. In September, the House reversed its position, authorizing $40 million for rat control in 1968 and 1969 [101]. 

An example of the efficacy of federal rodent control projects was the management of an epizootic of FBT in New Orleans. After an eleven-year hiatus, two human cases of FBT were presented in the city in 1971. An investigation revealed a large epizootic of FBT transpiring amongst the dockyard rats of New Orleans; 41% were found to be seropositive for FBT, compared to 7.3% during 1959–1970. After the failure of control measures implemented by local authorities, a combined federal-local program of insecticide application, followed by rodenticide treatment, then a second round of insecticide was successful in eliminating the rodent and flea populations and preventing further spillover of FBT into the human population [29].

From 1969 until the mid-1980s, a national urban rat control program was in place, first directed by the US Dept of Housing, Education, and Welfare, and later by the CDC. By 1978, 93 communities had received funds for improved sanitation, elimination of harborage, code enforcement, and rat extermination; 51,200 blocks were improved and over 23,000 blocks were rat-free, where 5.8 million persons lived [103]. In 1982, it was reported that rodent abatement programs had benefited nine million people living in 61,000 city blocks [104]. Southern cities that received funding for deratization included Little Rock, Pine Bluff (Arkansas), Houston, New Orleans, Mobile, Tuscaloosa, Pensacola, Miami, Atlanta, DeKalb Co. (Georgia), and Memphis [105]. By the 1970s, resistance of rats to warfarin was present in some areas of the USA [66], but newer, more potent anticoagulant rodenticides were then available. With the introduction of the highly successful anticoagulant rodenticides, Compound 1080 was banned in the USA in 1972 because of its toxicity [106]. Many local health departments in the USA continue to operate rodent control programs, although the level of proactive rat abatement among these departments varies considerably. The emphasis of these programs has moved away from poisoning and trapping to Integrated Pest Management, which is centered primarily on the elimination of conditions that promote rodent infestation [107]. The older multi-dose anticoagulants (warfarin, chlorophacinone, and diphacinone), have been superseded by newer anticoagulants, such as bromadiolone and difethialone, which are lethal after the ingestion of a single dose and are effective against rodents that are resistant to the older products [108]. Public and staff education remain important components of rodent control programs. From 2005 to 2015, New York City’s “Rodent Academy” trained over 2000 public health workers from all over the USA in the principles of rat elimination. However, the modern practice of rodent control suffers from a paucity of rigorous scientific research [107]. 

The changing epidemiology of FBT in Louisiana illustrates the long-lasting benefits of the FBT control programs of the post-war period. In 1945, there were 423 cases of FBT in Louisiana. That number dropped to only two cases in the period 1960 to 1969, three cases in the 1970s, none in the 1980s, and one in the 1990s [109]. 

## 4. Two Immunization Campaigns Against Flea-Borne Typhus

In 1944, Lavaca Co. (Texas) had the highest incidence of FBT in the country, at 438 per 100,000 persons [110]. In 1945, due to the high incidence of FBT, the CDC and the Texas Dept of Health organized a study of the extent of rodent infection with typhus in the county and found that 94% of the urban establishments and 77% of the farms and semirural premises harbored seropositive rats [111]. In fact, cases of FBT in the county were equally split between urban and rural areas; thus, successful control activities in Lavaca Co. would need to encompass both areas. From October 1945 to September 1946, the USPHS carried out DDT dusting in rural Lavaca Co. and the towns of Moulton and Shiner, with Hallettsville left as a control. DDT dusting reduced rat infestation by *X. cheopis*, but not by *L. segnis*, *E. gallinacea*, *P. spinulosa*, or *O. bacoti* [112].

Meanwhile, Texas State Health Officer Dr. George Cox endeavored to test an FBT vaccine in a civilian population. Hallettsville in Lavaca Co. was selected for the immunization demonstration site, since it was not undergoing DDT dusting. The rationale to perform a vaccine trial against FBT was based on the success of the epidemic typhus vaccine developed by Herald Cox of the Rocky Mountain Laboratory of the USPHS; millions of Allied troops received the vaccine during World War II, and not a single typhus fatality was recorded [113]. On 23 February 1945, the *Lavaca County Tribune* ran the frontpage story “People Here to Receive Free Typhus Vaccine Shots,” advising every man, woman, and child to report to City Hall for immunization. On March 6, 1945, the newspaper, under the title “Help Combat Typhus,” published a case report form requesting individuals with a history of typhus to convey this information to local authorities. It concluded: “Your help and cooperation will be greatly appreciated by all who are interested in whipping typhus fever and rats in Lavaca County.” By the time the program ended on March 16, 2100 people received the vaccine. Dr. Carl T. Dufner, Lavaca Co. health officer, and Dr. Harvey Renger, Hallettsville health officer, administered all of the immunizations in nine days. The completion of the program was feted with a ceremony: “An impressive program on the Courthouse Square… marked the successful conclusion of the typhus program here. It is considered that the program undertaken here has written a new chapter of medical science… The program was held in recognition of the voluntary response to this call in such a wholehearted manner. Dr. Cox addressed the public, as did Judge Paul Fertsch, Mayor Taxler, and other local officials. The high school Lion’s Club band rendered a number of selections. The stores closed at 2 pm for the occasion” [114,115]. According to one reference, the vaccine given in Hallettsville was specifically against murine typhus [56]. The source of the vaccine was reported to be the laboratories of Herald Cox (then at Lederle Laboratories) and M. Ruiz Casteñada of the Hospital Generale in Mexico City [116]. No further information on the vaccine nor the genesis or results of this trial can be located. 

Another immunization program against FBT was conducted by J. Wilfrid Davis and colleagues of the Baltimore City Health Dept after an outbreak of six cases occurred amongst the residents of a group of rat-infested row houses in 1946. An investigation found that 19% of 101 rats trapped in the houses were seropositive, as were 9% of 101 residents. The City Health Dept immunized 216 of the 328 persons living in the row houses with an unspecified vaccine against FBT. A DDT dusting and rat proofing/extermination program was also performed [117]. No results of this trial were published. 

## 5. Advances in Diagnostic Testing for Flea-Borne Typhus 

The first serologic test for FBT was the Weil-Felix test (WFt), which had been developed in 1915 for the diagnosis of epidemic typhus, but in the 1920s was found to be cross-reactive for the detection of FBT [118]. The WFt was based on the curious observation that the sera of patients with epidemic typhus would agglutinate a certain strain (OX19) of *Proteus vulgaris* [119]. The test was extended to other rickettsial infections, but care had to be taken to use specific strains of *Proteus* to provide the appropriate antigen or the specificity of the test was adversely affected. The OX2 and the OXK strains were more specific for the diagnosis of RMSF and scrub typhus, respectively. However, overall, the WFt suffers from a lack of specificity [120]. 

In FBT, the WFt may be positive as early as five days after the onset of illness, but it may take as long as 10–14 days to develop sufficient agglutinins [121,122]; the WFt titer will rise up to the time of fever crisis and positivity may persist for nine months [121]. A problem of the WFt was its cross-reaction in typhoid and paratyphoid. The practice of the Texas State Laboratory in the 1930s was to test specimens from febrile patients with a blood culture and agglutinins for brucellosis, typhus, typhoid, and tularemia. In typhoid, the blood culture may also be positive. The specificity of the WFt was enhanced if a high titer (≥1:160) was measured or if a second sample drawn days later showed a higher titer [122,123]. However, the subsequent development of complement fixation (CF) techniques rendered the WFt obsolete due to deficiencies in both sensitivity and specificity [124]. Following the peak years of FBT in the USA, it was likely over-reported because of misinterpretation of low titer Weil-Felix tests [125]. 

The CF serologic test for FBT was first developed by Ruiz Casteñada in 1936. The rickettsial antigen source was peritoneal washings obtained from X-irradiated, typhus-infected rats, but it was difficult to obtain large quantities of antigen [126]. In 1938, Herald Cox of the Rocky Mountain Laboratory discovered that various rickettsial organisms could be grown in chicken egg yolk sacs [127]. This permitted the production of large quantities of *R. typhi* antigens, which, in turn, led to the practical application of CF testing by Ida Bengtson and Norman Topping of the National Institutes of Health (NIH) in 1941 (Figure 8). The CF test proved superior to the WFt by four criteria. First, the CF test had improved sensitivity. In 322 cases studied by Bengtson, about 24% gave positive results with the CF test when the WFt was negative. Second, in tests with FBT-infected laboratory rats, Brigham and Bengtson found that the WFt positivity typically disappeared by the 20th day, whereas for CF testing, titers persisted over 10-weeks and in some cases out to six months. Thus, CF testing was superior to the WFt for epizootiologic assessment during FBT control projects [128]. Third, even low titers were significant in the CF test. Fourth, the CF technique was able to differentiate RMSF from FBT, an important advantage for areas in which the two infections overlapped. By the end of the 1940s, the CF test was beginning to supplant the WFt as the diagnostic technique of choice for rickettsioses [129,130].

In the 1960s, it was apparent that reliance on the WFt was also thwarting the ability of clinicians to make appropriate diagnoses of FBT and to make accurate epidemiologic assessments of the number of cases of FBT. For example, in Virginia, prior to 1970 the number of cases of FBT may have been underestimated due to confusion with RMSF, because both infections would give a positive WFt. In 1970, the Virginia State Health Dept Laboratory would reflexively perform the CF test if there were a positive WFt to help differentiate between the two infections [9]. 

However, CF tests have a number of limitations. The preparation of specific complement-fixing rickettsial antigens is expensive and requires a high degree of technical skill [131]. The tests are time-consuming and labor intensive. In the late 1940s, only a few laboratories had the expertise and materials to properly perform the test, including the NIH, the Rocky Mountain Laboratory, and the Division of Viral and Rickettsial Diseases of the US Army Medical School. Physicians sent samples to the aforementioned laboratories, but it usually required 10-days to three weeks to obtain the results [130]. Additionally, the CF test can be falsely negative if there are anti-complementary substances, such as IgG aggregates, present in the serum [132]. The CF test also has lower sensitivity compared to current serologic methods [133]. 

In 1976, Robert Philip, Elizabeth Casper, Richard Ormsbee, Marius Peacock, and Willy Burgdorfer of the NIH introduced the indirect fluorescent antibody (IFA) test, which is now the gold standard for the serodiagnosis of rickettsioses (Figure 8) [134]. In this technique, organisms are affixed to slides and the slides are treated with serial dilutions of the patient’s serum. Antibodies of the patient that are bound to the organisms on the slide are detected by fluorescein-labeled antibodies to human immunoglobulins. However, more than decade passed before the IFA test became readily available; the CF and Weil-Felix tests were more accessible than the IFA assay into the mid-1980s [135]. One deficiency of the IFA test is that antigens of *R. felis* and *R. typhi* and their respective human antibodies are mutually cross-reactive [136]. Thus, the current commercially available IFA serological assays cannot differentiate infection due to *R. typhi* from that of *R. felis*.

The IFA requires a fluorescence microscope, so there have been efforts to develop simpler latex agglutination and enzyme-linked immunosorbent assay-based diagnostic tests for FBT, but these have lower sensitivity than the IFA method or have not been sufficiently validated on clinical samples [137,138] and thus have not achieved widespread use. The polymerase chain reaction (PCR) proved to be a very sensitive tool to detect *R. typhi* and *R. felis* in fleas for epidemiologic studies [139]. However, PCR and other molecular diagnostic techniques, such as loop-mediated isothermal amplification (LAMP), have inadequate sensitivity for the clinical diagnosis of FBT from blood specimens because FBT has a low pathogen blood density (<100 organisms/mL of blood) compared to scrub typhus or RMSF [138,140]. For a summary of the diagnostic methods for FBT, see Table 2.

## 6. Nomenclature of Flea-Borne Typhus and Its Etiological Agent

Flea-borne typhus was variously termed typhus, Mexican typhus, Brill’s disease, or endemic typhus in the 1920s and 1930s. With the discovery of a rodent reservoir, Mooser named the disease murine typhus in 1932 [145]. By the 1940s, murine typhus was used by most investigators, with endemic typhus still being applied in some papers. However, endemic typhus is not an appropriate name for a disease due to *R. typhi* and *R. felis* because there is also a strain of *R. prowasekii* in flying squirrels (*Glaucomys volans*) that is also endemic in North America. Murine typhus is also no longer an accurate descriptor due to the lesser role of rodents in the current epidemiology. Recently, the term “typhus group rickettsioses” was used by investigators from the Texas Dept of State Health Services [146]. However, this is not scientifically accurate because *R. felis* does not taxonomically belong to the Typhus Group of rickettsiae [147]. The terms flea-borne spotted fever or cat flea typhus have been applied for the disease specifically due to *Rickettsia felis* [148]. From a practical clinical perspective, due to the inability to serologically differentiate the infections caused by *R. typhi* and *R. felis,* the term flea-borne typhus is best used to encompass both infections [149]. Due to current limitations of routine IFA testing, the term FBT would erroneously misclassify illnesses due to *R. prowasekii* (either an imported case or arising from a flying squirrel), but these are uncommon in the USA. 

Parasitologist John L. Todd and pathologist S. Burt Wolbach originally proposed the name *Dermacentroxenus typhi* for the organism of “Mexican typhus” [150]. However, the epithet *Dermacentroxenus* was later abandoned in favor of *Rickettsia*, to recognize Howard Ricketts, who had discovered the causative organisms of both RMSF and epidemic typhus [151]. Microbiologist José L. Monteiro named the organism *R. mooseri* in 1931 [152] to honor rickettsiologist Hermann Mooser, who had studied the pathogenesis of FBT in the 1920s and differentiated it from epidemic typhus [153]. Cornelius Philip of the US Army Medical School proposed the name *R. typhi* in 1943 [154] and this has been the valid name since 1980 [155]. 

## 7. The Treatment of Flea-Borne Typhus

In the pre-antibiotic era, there were no effective treatments for any rickettsial infection. Dr. Charles Reece of Austin, TX, stated in 1934: “The treatment [of FBT] is that of any acute infectious disease. Absolute rest in bed, careful nursing, sufficient nourishment, plenty of fluids, and treatment of symptoms as they appear, are about all that can be said. Opiates are usually necessary for headaches and nervous symptoms” [121]. While sulfa drugs entered the American pharmacopeia in the 1930s, this class of drugs had a deleterious effect in the treatment of rickettsial infection [156] and their avoidance for FBT was recommended in the 1940s [157].

In 1944, Grieff and Pinkerton reported that penicillin inhibited the growth of *R. prowasekii* in the yolk sac of chicken eggs [158]. However, clinically, penicillin has never been shown to be efficacious in any rickettsiosis. Penicillin given to patients with epidemic typhus in Italy and Egypt was ineffective [159]. In 1949, Diaz-Rivera and coworkers found that penicillin failed to alter the course of the illness in six patients with FBT [160]. However, penicillin was useful for the treatment of pyogenic secondary infection in FBT patients [157].

Thus, despite the introduction of sulfa drugs and penicillin, the treatment of FBT in the mid-1940s remained supportive only. Physicians Byron Stuart and Roscoe Pullen of Charity Hospital in New Orleans summarized the course of 180 FBT patients and prescribed “strict bed rest, good nursing care, maintenance of fluid balance and a liquid or soft nourishing diet with supplementary vitamins… Occasionally a stomach tube was necessary… to administer the necessary fluid and nutritional requirements. Digitalis was administered twice in cases showing… cardiac failure. Constipation was best controlled by using enemas, whereas adequate nursing care aided in preventing bed sores. Codeine and aspirin were frequently necessary for the control of the headache; however, in many instances only minimal relief was obtained. Temperature-reducing measures such as alcohol and coldwater sponge baths were employed freely. Quinine… was given to several of the patients with only transient effects on the [fever]” [157].

The first agent with activity against rickettsial pathogens was para-aminobenzoic acid (PABA). However, PABA had a low potency and required frequent dosing (every two hours) and therefore did not achieve widespread use [159]. In 1946, Paul Smith of the US Army Air Corps found that FBT patients treated with PABA defervesced several days sooner than controls [161]. Two other research teams also successfully treated FBT with PABA in the mid and late 1940s [160,162].

However, enthusiasm for PABA quickly evaporated with the discovery of chloramphenicol and the tetracyclines. Chloramphenicol was isolated in 1947 from the soil bacterium *Streptomyces venezuelae* by a team directed by Yale botanist Paul Burkholder (Figure 9) [163,164]; it was shown to be active against *R. prowasekii* infection of chick embryos, embryonated eggs, and mice by Joseph Smadel and Elizabeth Jackson of the US Army Medical Dept [165]. In 1948, Herbert Ley Jr., Theodore Woodward, and Smadel of the US Army Medical Research Unit treated three FBT patients with chloramphenicol and found that the patients defervesced in about 45 h after starting treatment (Figure 10). In four untreated FBT patients, the fever lasted for an average of 14.7 days [166].

Chlortetracycline, the first tetracycline, was discovered by Benjamin M. Duggar, a 71-year old retired botany professor (Figure 9). Duggar was renowned for his knowledge of soil fungi and possessed an extensive collection of soil samples that had been obtained from around the globe [167]. In 1944, Duggar was hired as a consultant to Lederle Laboratories and was tasked with screening soil samples to find a safer antibiotic than streptomycin for tuberculosis. In 1945, after screening 3400 soil organisms for antimicrobial activity, he finally hit pay dirt [168]. In a soil sample from the University of Missouri campus, Duggar discovered a bacterium, *Streptomyces aureofaciens,* which produced a gold-hued substance that exhibited antimicrobial properties. Further testing revealed it to be a novel, orally active antibiotic (chlortetracycline) with a broad spectrum of activity similar to that of chloramphenicol [169]. Large-scale production of chlortetracycline began in December 1948, [168]. A second tetracycline of greater aqueous solubility and potency, oxytetracycline, was isolated from *Streptomyces rimosus* by Alexander Finlay and coworkers of the Pfizer Company in 1950 [170].

Meanwhile, in 1948, Drs. Vernon Knight and Walsh McDermott from Cornell University Medical College and Francisco Ruiz-Sánchez and Amado Ruiz-Sánchez from the University of Guadalajara were investigating the treatment of typhoid fever with chlortetracycline in Mexico (Figure 10). Flea-borne typhus was common in Guadalajara, so they tried treating 11 FBT patients with chlortetracycline and found that headache and gastrointestinal symptoms disappeared overnight and the rash resolved in two to three days. The average duration of fever after the start of therapy was 1.7 days [171]. Chlortetracycline also reduced the period of prostration from 17 days to 10 days and the 50-day period of convalescence by more than half [33]. Most importantly however, the tetracyclines greatly reduced the mortality rates of the life-threatening rickettsioses, epidemic typhus, RMSF, and scrub typhus. This watershed event in the history of medicine was celebrated at the first Howard Taylor Ricketts lecture at the University of Chicago in 1950 [130]. The lecture was presented by Dr. Russel Wilder, and then a pathologist at the Mayo Clinic, who had been Ricketts’ research associate some forty years prior when Ricketts died in Mexico investigating the etiology of epidemic typhus. The title of Wilder’s lecture was “The Rickettsial Diseases: Discovery and Conquest.” In his lecture, he asserted that “This discovery of a cure for the rickettsial diseases represents the final chapter of an epic” [172]. Of course, much remained to be done in regards to the epidemiology, pathogenesis, and prevention of these infections, but clearly, with the availability of efficacious antibiotics against rickettsioses, a great medical advance had been achieved [173].

However, the chemical structures of both chlortetracycline and oxytetracycline were unknown, and in a time before nuclear magnetic resonance spectroscopy and mass spectrometry, this obstacle was daunting. However, in 1953, a team of Pfizer chemists, directed by Karl Brunings, and Harvard University chemist Robert Woodward, deduced the structure of oxytetracycline, using only the elemental analysis, data from chemical degradation and modification studies, and infrared and ultraviolet spectroscopies (Figure 11 and Figure 12) [167,176]. This discovery of the structure of oxytetracycline paved the way for the synthesis of oxytetracycline analogs with improved stability and pharmacokinetic and pharmacodynamic properties.

In 1958, a team of chemists at Pfizer, led by Charles Stephens, starting with a derivative of oxytetracycline, synthesized doxycycline, a drug with improved oral bioavailability, enhanced tissue penetration, and a longer half-life compared to the natural tetracyclines (Figure 12) [168]. Doxycycline became the drug of choice for the treatment of rickettsioses after its approval in 1967 because of its easier dosing schedule (twice daily) than tetracycline (four times a day) [177]. During treatment of FBT with doxycycline, the fever typically remits within 48 to 72-h [178]. Even for children less than eight years old, doxycycline is the preferred treatment, despite the risk of dental staining [179].

By 1950, chloramphenicol was suspected of triggering a potentially fatal aplastic anemia [180]. Reports of aplastic anemia associated with chloramphenicol continued through the 1950s and 1960s. In 1963, the California State Senate asked the California State Dept of Health to investigate the matter. In 1969, their study concluded: “The risk of fatal aplastic anemia is sufficiently remote that many physicians can treat thousands of patients without encountering a fatality. Yet to assume a probable risk 13 times the normal risk appears totally unwarranted… if a safer alternative drug is available” [181]. Thus, the tetracyclines essentially became the only available safe drugs to treat rickettsial infections.

In the 70 years since the introduction of the tetracyclines, other antibiotics known to be effective against intracellular pathogens, such as fluoroquinolones and macrolides, have been pressed into service to treat FBT. However, ciprofloxacin failures have been reported [182]. Azithromycin has been used to treat FBT during pregnancy to avoid the potential adverse effects of tetracyclines on the developing fetus [183]. However, a recent clinical trial clearly demonstrated the superiority of doxycycline over azithromycin in the treatment of FBT [184]. A recent re-assessment of doxycycline concluded that it is safe in early pregnancy, possibly throughout pregnancy, and for children [177].

The widespread use of tetracyclines and chloramphenicol in the United States in the 1950s probably caused a decrease in the number of apparent cases, because, if effective antibiotics were administered early in the course of FBT, the appearance of an antibody response may have been prevented or delayed [6,7], with the loss of protective immunity [7].

## 8. Another Organism Causing Flea-Borne Typhus Is Discovered

In 1990, an organism resembling *R. typhi*, originally termed the ELB agent, was discovered in the cytoplasm of midgut cells of cat fleas collected from feral cats by Jean Adams and Edward Schmidtmann of the USDA and Abdu Farhang Azad of the University of Maryland [185,186]. This organism would later be named *R. felis*. In 1992, Williams and coworkers reported that cat fleas obtained from opossums in Los Angeles also harbored this microbe [187]. Dogs and cats were also found to be reservoirs [188,189]. In 1994, using PCR, *R. felis* was found to be a cause of FBT in a patient from Texas (Nueces Co.) [190]. By 2009, cases of human FBT due to *R. felis* were also recognized in 11 other countries [191]. Infection due to *R. felis* appears to be milder than that from *R. typhi* [192]. However, because serologic methods to differentiate *R. felis* from *R. typhi* are not readily available, there are limited clinical comparisons between the illnesses caused by the two pathogens. 

The widespread distribution of *R. felis* is primarily due to the cosmopolitan nature of the cat flea. *Rickettsia felis* has been found in more than 40 other species of fleas, ticks, mosquitoes, and mites [193]. However, the cat flea is the only proven vector [194]. In the absence of readily available diagnostic techniques to differentiate the infections due to *R. typhi* and *R. felis*, the relative contribution of these two pathogens to the burden of FBT in different geographic areas has not been established. 

## 9. Epidemiologic Trends in Current High-Incidence Areas (Texas, California, Hawaii). General Considerations 

With the new insecticides and rodenticides, improved living standards, and urban renewal, the tide had turned against FBT in the American South by the mid-1950s. At that time, with the availability of specific CF serologic tests to confirm the diagnosis, the disease now appeared to be clinically over-diagnosed in areas of sporadic occurrence [195]. In 1979, 70 cases were reported in the US, with 59 occurring in Texas [133]. A decade later, only 41 cases of FBT were recorded for the entire nation [196]. With its waning national significance, in 1987, the Council of State and Territorial Epidemiologists removed FBT from the list of nationally notifiable conditions [8]. However, despite the success against the rodent-borne disease, FBT was found to be exploiting new ecological niches.

As the offensive against rats and their fleas was progressing apace, there were early indications that the epidemiology of FBT in mainland USA was evolving. In the 1950–1960s, there was a shift in the location of FBT cases in the Los Angeles area from the central and south–central districts of the city to the foothills of eastern Los Angeles and Orange counties. In 1951, an outbreak of 33 cases of FBT in Altadena, northeast of Los Angeles, led to an investigation that yielded only a single seropositive rat and failed to find Oriental rat fleas. In 1967, a study conducted in Orange Co. (California) by William H. Adams and colleagues of the California State Dept of Health determined that 11% of 75 opossums were seropositive and *R. typhi* was isolated from the spleen of an opossum trapped at the home of a FBT victim [197].

Physicians John Chapman and A.A. Chapman of Sweetwater, Texas, had originally proposed opossum *(Didelphis virginiana)* as a possible reservoir in 1935, after they diagnosed FBT in two possum hunters with flea exposure [198]. In 1936, Brigham verified that opossums were susceptible to FBT [199]. The potential significance of opossums as reservoirs of FBT was then ignored for thirty years. Subsequently, the presence of an opossum reservoir was demonstrated in 1998 in Corpus Christi, TX; 8% and 22% of opossum sera were reactive for *R. typhi* and *R. felis*, respectively. The study also revealed a close geographic association between human FBT cases in Corpus Christi and opossum contact; six of 19 patients lived within the minimum home range of a seropositive opossum while five more lived within the maximum home range [200]. The prominent role of the opossum in the current epidemiology of FBT in Texas was again demonstrated by an outbreak investigation in Austin/Travis Co. in 2008 that involved 33 patients. The outbreak was unexpected because only four cases of FBT had been diagnosed in Travis Co. since the eradication efforts of the late 1940s; two of these cases occurred in 2007, suggesting that they may have been sentinels for the 2008 outbreak. A serologic investigation was conducted on the peri-domestic animals located near the homes of 21 patients. The prevalence of rickettsial seropositivity for the 16 opossums, nine dogs, and 17 cats on these premises were 71%, 44%, and 18%, respectively; nine raccoons and four rats were seronegative. Cat fleas comprised 84% of the potential arthropod vectors that were captured at these sites [201]. 

Opossums are a problematic reservoir for FBT because they may live in close proximity to humans and can thrive in urban, suburban, and rural areas. Opossums readily feed on garbage and pet food, putting them in proximity to pets. The significance of opossums in the epidemiology of FBT is amplified by the huge number of fleas that they are able to host. In a study of 259 opossums trapped in Orange County (California), the average flea burden was 91 (range 0–726), or roughly nine-times that of a brown rat. Other backyard wildlife, such as skunks and raccoons, harbor few fleas [202]. 

## 10. Flea-Borne Typhus in Texas, 1940s to Present 

Throughout 1930s and up until 1946, Texas consistently ranked third behind Georgia and Alabama in the total number of FBT cases (Table 3) [203]. Cases of FBT in Texas peaked in 1944 at 1740 [204]. Since 1946, the Annual Summary of Notifiable Diseases published by the Texas Dept of Health has included FBT. The reported cases of FBT in Texas dropped rapidly with the implementation of the post-war rodent and flea extermination programs; by 1952, there were less than 100 cases [203]. 

In 1951 a study was undertaken by the Texas Dept of Health to compare the 1946 and 1951 seroprevalence rates for FBT in the rat population at 50 urban and rural sites [204]. Due to a shortage of rodent control personnel in rural Texas, 2060 high school students were trained in the practice of rodent mitigation during a four-day course that addressed: rat proofing, poisoning, and trapping; rodent-borne diseases; DDT use; and the collection and processing of rat blood. The involved high schools were fully supportive and were used as sites for blood collection from the rats (Figure 13).

In 1946, in the eight Texas survey counties (Bexar, Dallas, El Paso, Galveston, Jefferson, Karnes, Nueces, Tarrant), of 495 premises sampled, 42% had seropositive rats, versus 14% of 580 premises in 1951. Karnes Co. had the highest rat seropositivity rate in 1946, at 75%; in 1951, the rate was only 8.7%. Cameron Co. also managed a successful rural program, with DDT dusting and rat extermination on all infested farms. In 1947, 34% of the farms had seropositive rats. In 1951, this was only 6% [204]. 

Corpus Christi (CC) was an exemplar of successful urban typhus control. In 1946, 87% of the food establishments in CC were rat-infested; in 1951, this had been reduced to 11%. Nineteen percent of tested premises had seropositive rats in 1946 versus zero in 1951. Anti-typhus measures implemented in CC from 1946 to 1951 included: (1) semi-annual DDT dusting of rat-infested businesses and annual dusting of residential areas; (2) rat proofing with an emphasis on food handling establishments (cafes and grocery stores were required to be rodent-free to receive business permits); (3) rat extermination in all business and residential premises; (4) anti-rodent sanitation practices in residential and business areas; (5) municipal garbage disposal by sanitary landfill; and (6) adoption of ordinances regulating livestock and animals. Within CC, these measures reduced rodent and flea populations to very low levels. However, such intensive control measures were not implemented in the rural areas and smaller towns of Nueces County. The county reported 59 FBT cases in 1945, but only nine in 1950, which is an 85% reduction. To eliminate the remaining 15% required effective typhus control programs for rural Nueces County [204]. 

In contrast to the progress in CC, San Antonio (Bexar Co.), after being a demonstration site for the effectiveness of DDT in 1945, had allowed its FBT mitigation program to languish. In 1946, 80% of tested premises in San Antonio were rat-infested; in 1951, it was 78%. Of the 93 premises from which rat sera were tested in 1946, 40% were positive for typhus-infected rats, versus 39% of 75 inspected premises in 1951. The problem was that San Antonio had neglected the poorer residential districts of the city. One extensive cycle of DDT dusting in both businesses and residential premises was conducted in 1945–46. Dusting after 1946 was done primarily in businesses, with spot dusting done in the immediate areas in which typhus cases had been identified. Rat control measures that had been carried out included rat proofing in the central business district, rat poisoning in the municipal buildings and grounds, inspection of food establishments, and application of rodenticides by commercial exterminators. A large portion of the rat infestation was found in the southwestern sector of the city adjacent to the central business district. This area consisted of small, crowded, dilapidated homes, with interspersed taverns, groceries, and cafes. On the premises where rats had been trapped in 1946, 40% yielded rats with *X. cheopis*, increasing to 45% in 1951. However, the *X. cheopis* flea index (fleas/rat) did decrease, from eight per rat in 1946, to only two per rat in 1951. This was likely due to prior DDT dusting. Due to the drop in the flea index, cases of FBT did decrease in San Antonio, from 37 in 1946 to one in 1950. As a result of the survey findings, extensive DDT dusting of the rat-infested premises throughout the impoverished area of the city was carried out. Rat extermination with warfarin was also implemented in premises with heavy infestation. The experience in San Antonio demonstrated that sporadic DDT dusting can reduce the prevalence of typhus in humans, but multi-modality rat control and DDT dusting are necessary to reduce prevalence in the rat population [204]. 

Up through 1960, the number of cases of FBT in Texas continued to fall, averaging 20 cases/year [200]. At that point, Texas was accounting for about 80% of American FBT cases. Due to the persistence of FBT in the state, in 1969 the CDC dispatched an officer of the Epidemic Intelligence Service to investigate the situation. Of the 28 Texas cases in 1969, 23 were reported from the southern counties of Hidalgo, Nueces, and Cameron. Only 10 cases had definite animal exposure: six with cats; two with opossums; and two with rats [206]. Thus, in Texas in the late 1960s a transition from a rat flea-borne disease to cat flea-borne disease was already transpiring. During 1975–1981, there was an average of 63.4 cases/yr in the USA, with 78% originating in Texas [7]. From 1980 through 1984, there were 200 cases of FBT reported in Texas, with four counties dominating: Nueces (57 cases) and Kleberg (9 cases) along the Gulf Coast Bend and Hidalgo (39 cases) and Cameron (22 cases) in the Lower Rio Grande Valley. In Nueces and Hidalgo Counties, the incidence was 4.2 and 2.7 per 100,000 residents, respectively. In this series of 200 patients in the post-tetracycline era, the case-fatality rate was 1% [133]. In a study of opossums conducted in Corpus Christi in 1984, 60% were seropositive for *R. typhi*. FBT persists in South Texas because the temperature rarely gets below 7.2 °C, which permits more rapid flea life cycles [207].

By the mid-1980s, the ecology of FBT in South Texas had been transformed. Martin Schriefer and coworkers at the University of Maryland assessed for the presence of *R. typhi* and *R. felis* in opossums, cats, dogs, rats, and their fleas in Corpus Christi in 1984–1986 and in 1991. Rats and dogs were seronegative, but 31% of the opossums and 4% of the cats were positive. None of the trapped rats harbored fleas and their mites were negative for *R. typhi*. Of the cat fleas that were captured on the opossums, cats, and dogs, 5.9, 5.7, and 11.1% were positive for *R. typhi*, respectively [208].

In a study done in 2005–2006 on 513 children residing in Nueces Co. the seroprevalence of *R. typhi* was 9–14%, depending on the age group [209]. Due to the high incidence of FBT in Nueces Co. and the role of the opossum as its reservoir, the city of Corpus Christi is unique in that it maintains a budgeted staff position and a vehicle dedicated to opossum control [210].

Flea-borne typhus was common in Galveston in the mid-1940s [30], but after a control program was implemented in the late 1940s, it was free from the infection for six decades. However, in 2013, twelve cases were diagnosed in the city. An assessment of 500 serum samples from an outpatient clinic showed a 1.6% seropositivity rate for FBT [211]. This indicates that despite the diagnosis of 12 cases, exposure to FBT in Galveston was uncommon. Subsequently, it was found that 8 of 12 locally caught opossums were seropositive for *R. typhi*. All of the opossums were infested with fleas; 7 of 12 opossums harbored fleas with evidence of *R. typhi* infection [212]. Thus, even in a seaport such as Galveston, opossums had likely replaced rats as the principal FBT reservoir.

During 2003–2013, a total of 1762 Typhus Group Rickettsioses (TGR) cases (770 confirmed and 992 probable) were reported to the Texas Dept of State Health Services (TDSHS). The term TGR is used instead of FBT because the IFA test is unable to differentiate FBT from epidemic typhus, but these cases are unlikely to be the latter. The TDSHS epidemiologists were unable to define the reasons for the uptick in the number of TGR in TX in the last decade; they doubted that greater case ascertainment or reporting was the cause. During 2003–2013, the number of annual cases varied from 27 in 2003 to 222 in 2013. An average of 102 cases were reported annually during 2003–2007, less than half the average number (209) reported during 2008–2013. Overall, FBT cases peaked in June and July; however, in south Texas (<28° N), peaks occurred in both June/July and the other in December/January. The reason for this bimodal distribution of cases is unknown. Geographic expansion of FBT within Texas also occurred during the 2003–2013 period. In 2003, cases were reported from nine counties in south Texas, but by 2013, 41 counties were affected [146]. Furthermore, cases of FBT may still be undercounted. Rickettsial infections are reportable in Texas, but in one study in Houston, only 48% of the local pediatric FBT patients were listed in the state’s communicable disease surveillance database [213], indicating significant gaps in the reporting system. Due to the increasing geographic distribution and numbers of cases of FBT in Texas, on November 30, 2017, the TDSHS issued a “Typhus Health Alert,” requesting that clinicians intensify their suspicion for FBT [214]. The advisory provided information on clinical aspects, epidemiology, reporting procedures, and laboratory testing. Cases of FBT continue to rise in Texas; in 2018, there were 738 cases, the most in the state since 1946 (Table 3) [205]. 

## 11. Flea-Borne Typhus in California, 1915–2019

Maxcy recorded the first two cases of FBT in California, in San Francisco in 1915 and 1919. The first case in Los Angeles was seen in 1920, followed by four cases in 1921 [215]. There were 32 cases of a typhus-like illness in Los Angeles Co. between 1919 and 1924 that were likely flea-borne [216]. In 1924, an epidemic of pneumonic plague struck Los Angeles. To combat plague, 210,000 rats were trapped and extensive rat proofing measures were implemented in the city, likely decreasing the subsequent number of FBT cases [217]. From 1923 to 1928, only 23 cases were reported, from Los Angeles, San Diego, and San Bernardino counties. From 1919 to 1945, there were 362 cases in California, with a 4.7% mortality rate. Cases were not equally distributed over this period. There were only 22 cases from 1925 to 1935, or about two cases per year. However, from 1936 to 1941, there were 123 FBT cases, or an average of 20.5 cases per year. During the war years (1942–1945), the number of cases swelled to 40.1 per year [218]. Nevertheless, there were still less cases in California from 1919 to 1945 than in any of the Southern states [4,217,219]. During 1919 to 1945, three counties accounted for 88% of California’s cases: Los Angeles (67% of the total), San Diego (18%), and Orange (4%) [218]. One can speculate that the increase in cases during the war years was fostered by the increased population densities and poorer sanitary conditions in the military cities of Los Angeles and San Diego. 

There is little written on specific typhus control activities in California in the first four decades following World War II. In 1947, in order to capitalize on the newly available insecticide and rodenticide technologies, the California State Dept of Public Health established the Bureau of Vector Control. The new bureau consolidated the mosquito and rodent control activities previously conducted by other agencies and consisted of nine public health engineers and/or entomologists, six sanitarians, and 21 rodent control workers. Typhus was one of the nine priority vector-borne infections addressed by the Bureau [220]. In 1948, several California cities in the three high FBT incidence counties joined the National Urban Rat Control Campaign (vide supra) [62,63]. However, at that time, the incidence of rat seropositivity for typhus was low. In San Diego, Orange, and Los Angeles counties, only 5% of 163 rat sera tested in 1948–1950 were seropositive [221]. In 1950, the California State Dept of Public Health published the comprehensive monograph *Typhus Fever in California, 1916–1948, Inclusive* on the epidemiology of FBT in the state, which included practical information on rat proofing and the use of rodenticides and DDT [54].

Even though the rat control programs that had been implemented by the CDC were terminated in the mid-1980s, the health departments in Los Angeles, Riverside, San Bernardino, and San Diego counties continued rat and flea control programs due to concerns about plague [216]. Beginning in the early 1950s, a shift was observed in the distribution of FBT cases in California from urban areas to suburbia. The Oriental rat flea was not associated with these suburban cases [197]. In a study of the flea infestation of 1206 roof rats collected in suburban Los Angeles Co. in 1981–1982, no *X. cheopis* were found, indicating that the previous main vector flea species for humans was now uncommon in this area [222].

The presence of *R. felis* in cat fleas was discovered in 1990 [185] and the proclivity of cat fleas to feed on opossums was established soon thereafter [187,191,223]. In 1993, Sorvillo and coworkers investigated 33 cases of FBT in Los Angeles; 40% of 36 opossums and 90% of 10 cats trapped near human cases were seropositive for *R. typhi*, whereas only 6% of 36 black rats and none of the 35 brown rats were seropositive. The cat flea was the predominant flea species identified on the opossums; no *X. cheopis* were found. This study implicated cats in the epidemiology of FBT in California and established the opossum as an important reservoir and host of the cat flea [224]. 

The abundance of cat fleas on opossums was significant because this flea is also the most common species to attack domestic dogs and cats, thereby providing a potential route for the transmission of FBT to humans [224]. In 2006, there was an outbreak of 21 cases west and south of Los Angeles, an area usually free of FBT. It was speculated that FBT invaded this virgin territory because of opossum relocation programs and the migrations of feral cats [225]. Thus, a suburban cycle involving cats, opossums, and cat fleas was responsible for the human cases of FBT in suburban Los Angeles [187,197,224].

Opossums are not native to California and were originally introduced into the Los Angeles area in 1890; this population extended into Ventura Co. by 1924. Another population of opossums was brought into Santa Clara Co. in 1910 [226]. In 1924, an entrepreneur living in Tulare Co. brought opossums into the state to raise for their fur, which at the time was used as an inexpensive trim for clothing. However, the demand for possum pelts collapsed, and the frustrated furrier released his unmarketable marsupials into the wild [227]. Opossums now inhabit the Pacific coastal states from San Diego up to Canada. In California, they are present west of the San Joaquin and Sacramento Valleys [228]. It is curious that, despite the widespread range of the opossums and the cat flea in California, FBT primarily occurs in a geographically restricted area focused on Los Angeles and adjacent Orange Counties. From 2001 through to 2015, there were 479 cases of FBT in Los Angeles Co. and 1142 in Orange Co., but only 20 cases elsewhere in the state [229]. 

There were few reported cases of FBT in California between 1946 and 1995. Between 1984 and 1994, the Los Angeles Co. Dept of Health Services received 75 reports of FBT [230]. However, human cases started increasing again in the 1990s, and by 1996 there were again more than 10 cases of FBT annually in California [229]. In 1998, the California legislature passed the Hayden Act to reduce overcrowding and euthanasia at animal shelters. Under the previous law, unwanted cats impounded by animal shelters could be euthanized after three-days. The Hayden Act increased the minimum duration of impoundment to six days, and required impounded animals to be released to animal rescue organizations under certain circumstances. Local jurisdictions, including in Los Angeles Co., passed additional ordinances to reduce euthanasia of impounded animals and to implement “no kill policies,” which included sterilization and relocation of feral cats (“trap-neuter-return” (TNR)). The Hayden law shifted the mission of animal control agencies from public health to animal welfare. As more animal control services were provided by contracted animal rescue agencies, the capability of animal control agencies to respond to outbreaks of zoonoses was degraded. For example, in a 2015 FBT outbreak in Pomona, public health personnel mobilized to confront the outbreak, but the local contract animal control agency declined to participate [216]. Furthermore, penalties against feeding wild or feral animals were reduced, which promoted increased feeding of feral cats and discouraged shelters from impounding unclaimed animals [231].

In the last decade, California has become ground zero in the conflict between public health authorities and TNR proponents. In 2014, Orange Co. (OC) Animal Care initiated a TNR program to address the overpopulation of feral cats as an alternative to euthanasia. However, the Orange Co. Mosquito and Vector Control District (OC Vector Control) opposed OC Animal Care’s TNR program because of the resurgence of FBT in the county. Orange Co. had been free of FBT from 1993, but the infection resurfaced in 2006. From 2006–2016, 22% (159/731) of California cases of FBT occurred in Orange Co. [232]. An investigation of 66 human FBT cases in the period 2012–2013 by OC Vector Control showed an association with stray or feral felines in 27% [202]. In 2014 OC Vector Control recommended that OC Animal Care conduct a California Environmental Quality Act review of their TNR program to assess potential health risks posed by the cat releases. OC Animal Care declined this recommendation and continued to advocate for their TNR program [233].

In California, each county is required to annually assemble a grand jury to evaluate local governance. In their 2014–2015 report, the OC Grand Jury questioned the ability of the OC Animal Care TNR program to reduce feline overpopulation and identified it as a possible contributor to the spread of zoonoses [233]. OC Animal Care disputed this assessment and countered that it was the previous 60-year-long practice of “trap and kill” that had promoted feral cat population growth. OC Animal Care claimed that seven to ten years of data would be necessary to properly evaluate its TNR program. Conversely, OC Vector Control maintains that for a TNR program to achieve long-term declines in feral cat populations, 75–95% of the cats in a feral colony must be sterilized to overcome feline fecundity. Since most TNR programs do not have the resources to undertake such extensive operations, they will fail to control cat populations [232]. 

In response to an FBT outbreak in Santa Ana in 2012, OC Vector Control placed traps to capture feral cats near the school of an FBT-infected child; however, the anti-feline crusade was short-lived. The felinophiles of California quickly mobilized to defend the strays. An article in *Animal People* pooh-poohed the significance of FBT and the potential role of strays in its transmission compared to housecats. University of California faculty member Deborah Ackerman opined that trapping feral cats was unlikely to be an effective typhus control measure because pet cats were more likely to be the important feline reservoir [234]. OC Vector Control and the Santa Ana Police Dept were flooded with calls from those who opposed the cat trapping. An online petition drive was mounted and a protest was planned. People even went to the school to trigger the traps before cats could be captured. Before the protest occurred, the county officials withdrew the traps without a single cat being bagged. The felinophile newsletter *Catster* called the OC Vector Control effort “ridiculous feral cat witch hunts” [235]. However, an analysis of FBT cases in Orange Co. revealed clustering in areas with high TNR cat density compared to less affected areas of the county [236]. 

Since 2010, cases of FBT have escalated in California. From 2001 to 2009, there was a total of 151 cases of FBT, or an average of 16.8 cases per year. From 2010 to 2018, there were 868 total cases, an average of 96.4 cases per year, a 574% increase in this decade compared to the prior. The year 2018 saw the most cases of FBT in California since records have been kept, with 172. From 2010–2018, 77% and 20% of cases occurred in Los Angeles and Orange Counties, respectively [229].

An investigation of an outbreak of five cases of FBT in a mobile home park in the San Gabriel Valley, Los Angeles Co., in 2015 highlights the issues of the current FBT situation in California: the involvement of feral cats and opossums; the high flea burden of opossums; the presence of *Rickettsia* in the flea population; and inadequate flea control for pets. Investigators captured three opossums hosting 487, 615, and 1087 cat fleas and 13 stray cats with 0 to 13 fleas/cat. Of the fleas on the opossums and the cats, 13% and 22% were positive for *R. felis* by PCR, respectively. None of the opossums were seropositive for *Rickettsia*, whereas 46% of cats were seropositive. Public health authorities removed the feral cats and opossums, provided flea-collars for pets, and outdoor feeding sources were removed; no additional cases of FBT occurred during a post-intervention surveillance period [237].

Nevertheless, rats and their fleas still play a role in the epidemiology of FBT in California. Serosurveys conducted from 1996 through 1998 of 259 brown rats in downtown Los Angeles revealed 26% seropositivity for *R. typhi* [238]. In 2008, Abramowicz and coworkers collected four rats in an area of urban Los Angeles with poor environmental sanitation. *R. felis* and *R. typhi* were detected in the tissues of four and three of these rats, respectively. DNA from *R. typhi* and *R. felis* were detected in *X. cheopis* from each rat with infection rates of 10% and 32% of the fleas, respectively. Thus, both *R. typhi* and *R. felis* are likely still present within the rat and flea populations of urban Los Angeles. While the presence of *R. felis* in opossums and cat fleas was widely recognized, this study demonstrated that this pathogen is also present in rats and *X. cheopis*. The co-infection of rats with both *R. typhi* and *R. felis* observed by Abramowicz and colleagues was a novel finding [239]. Co-infection of fleas with both rickettsial species has also been reported [240,241]. It is unknown if either pathogen has an advantage in terms of acquisition, persistence, or transmission [240,242]. However, Wiggers, Martin, and Bouyer have proposed that *R. felis* is less pathogenic than *R. typhi* because, despite its common occurrence in opossums and cat fleas, *R. felis* is less likely to cause human infection; furthermore, human body temperature is suboptimal for its growth [136]. In 2012, in southern California, a minimum infection rate of 1.7% for *R. typhi* was found in pools of fleas collected from opossums living in Orange and Los Angeles Counties. By contrast, fleas obtained from opossums inhabiting San Bernardino Co., an area with few human FBT cases, revealed no evidence of infection with *R. typhi* [242]. In 2016, Billeter and Metzger collected cat fleas from cats in Los Angeles Co. (an endemic area for human FBT) and in Sacramento and Contra Costa Counties (non-endemic areas). PCR confirmed the presence of *R. felis* in cat fleas from both the endemic and non-endemic areas; *R. typhi* was not detected. Because *R. felis* is widespread in cat flea populations in both FBT endemic and non-endemic areas, the investigators proposed that it is unlikely that *R. felis* is a major cause of human FBT in California [243].

On October 4, 2018, the Los Angeles Co. Dept of Public Health issued an alert stating that between July and September 2018 there were nine cases of FBT in persons living or working in downtown Los Angeles. Six of the patients reported a history of homelessness or living in interim housing [244]. The outbreak of FBT in downtown Los Angeles continued into March 2019, numbering 19 cases [245] and has been blamed on the noisome homeless encampments of the Skid Row district. The area bounded by 3rd, 7th, Spring, and Alameda Streets acquired the appellation “The Typhus Zone” [246]. In November 2018, a Los Angeles Deputy City Attorney was stricken with FBT, which she believed she contracted at Los Angeles City Hall, and she implored that the city take action to protect other employees. (She subsequently filed a five-million-dollar lawsuit against the city) [247]. Fumigation of several Los Angeles Police Dept facilities was conducted due to flea infestation. Vicki Curry, spokeswoman of the city of Los Angeles, issued the statement: “Last fall (2018) we directed multiple city departments to begin a coordinated and comprehensive effort to improve cleanliness and protect public health in the Civic Center, including City Hall and City Hall East. In addition to increased trash collection and cleanings, aggressive action has been taken to address pests both in the buildings and in the surrounding outside areas—including abatement treatments and the filling of 60 rodent burrows and 114 tree wells [248].” The office of Los Angeles Mayor Eric Garcetti declared that in October 2018 that an additional $300,000 was allocated to sanitation in the “Typhus Zone” [249]. 

There were efforts to politicize the 2018/2019 outbreak of FBT in Los Angeles Co. as a failure of governance by California Democrats. The Fox News Channel aired two stories on the FBT outbreak in Los Angeles, on 2/4/19 and 3/8/19, both on *Tucker Carlson Tonight* [250,251]. Carlson incorrectly called FBT “a medieval disease,” “a disease you read only about in novels,” and “a disease eradicated in the 1800s.” The Fox News Medical Correspondent Marc Siegel erroneously called FBT “an ancient scourge” [251]. On 3 June 2019, Fox News aired another story which confused typhoid with FBT [252]. The FBT outbreak in urban Los Angeles was also used to demonize the homeless and justify gentrification of the Skid Row district. However, in addition to the Skid Row outbreak, there were also 22 cases of FBT reported from Pasadena, ten miles from downtown Los Angeles, which did not involve homeless persons [253]. On 22 March 2019, the LA County Public Health Department sent a letter to the mayors of all county municipalities, detailing the local epidemiologic situation and recommended preventative measures [245].

The rat problem in Los Angeles pre-dated the 2018 typhus outbreak. In 2002, the *New York Times* published an article about rats in Beverly Hills, noting the attraction of “fruit trees, bird feeders, swimming pools, and dog-food bowls” to the vermin [254]. Also, the County of Los Angeles Dept of Public Health Vector Management Program had previously issued the bulletin *The Norway Rat in Downtown Los Angeles* [255]. While detailed epidemiologic studies of the recent Los Angeles cases have not yet been published, it is evident that, in the county, there are now both an urban cycle of FBT involving rats as the major host and a suburban cycle with opossums and pets/strays as hosts.

## 12. Flea-Borne Typhus in Hawaii 

Flea-borne typhus was first recognized in the state of Hawaii in Honolulu after the WFt was introduced into medical practice there in 1934 [256]. From 1934–1938, Honolulu accounted for 75% of the cases in the territory [257]. From 1937–1941, there were 202 cases in Honolulu [258]. In 1942, 12% of the rats in Honolulu (on Oahu island) were infected with FBT [259]. Honolulu has a warm, humid climate conducive to the proliferation of rat fleas [260] and had several attributes favorable for rodent infestation. In the residential districts, the stone walls, chicken coops, and lush vegetation provided ample rat harborage. A ready source of rat food was afforded by plentiful mango, coconut, and avocado trees that were often planted close to human dwellings. On nearby hillsides and vacant lots grew guava, cacti, and other fruit-bearing plants [257]. 

The number of annual cases of FBT in Hawaii peaked in 1944, at 186; 80% of cases occurred on Oahu, 15% on Maui, 4% on Kauai, and 1% on Hawaii. In 1950, the number of cases of FBT had dropped in the state to twelve as a result of DDT dusting programs and building codes that required rat-proof construction [261]. The usual flea species that vector FBT are present in Hawaii, including *X. cheopis*, *N. fasciatus*, *L. segnis*, *C. felis*, *Pulex irritans*, and *E. gallinacea* [262]. In post-World War II Honolulu, 60% of the brown rats of were infested by *X. cheopis*, with an average of 3.1 fleas/rat; 42% of black rats were infested, with 2.7 fleas/rat [263]. There are several unique ecological considerations that have affected the epizootiology and epidemiology of FBT in the Hawaii. Serologic studies have identified four rodent reservoirs: the Polynesian rat (*Ra. exulans*), black rat, brown rat, and the house mouse. However, the Polynesian rat is small (about half the size of *Ra. rattus*) [264] and so typically does not harbor many fleas [263], making it less significant as a reservoir compared to other rat species. The Indian mongoose (*Herpestes auropunctatus*) has also been proposed as a reservoir [241]. Rodent burrows in fields under cultivation provide a favorable microhabitat for rodent and flea survival. The dust in these burrows is often contaminated with flea feces that contain rickettsiae that may remain viable for prolonged periods. Additionally, a transition from sugar cane cultivation to corn in some areas of the state has also been favorable to rodent proliferation. The lower prevalence of FBT on the islands of Kauai and Hawaii is due to higher levels of rainfall [261]. Very high moisture levels inhibit the development of flea larvae in the soil and heavy rainfall washes the flea eggs and larvae away from the flea-infested host animals, which are a necessary source of the adult flea feces and infertile eggs that nourish the larvae [265]. In the Hawaiian Islands, more FBT cases occur in the drier leeward areas [261].

The epidemiologic data for FBT for the four counties of Hawaii from 1990 to 2016 are shown in Table 4. The Kihei zone of Maui, the driest area of the island [266], was recognized as a hyperendemic focus in 1972 [261]. From 1990 to 1999, there were 42 cases of FBT in state of Hawaii, with zero, six, four, and 32 cases from the counties of Hawaii, Honolulu, Kauai, and Maui, respectively. Thus, Maui Co., which consists of the islands of Maui, Lanai, Molokai, and Kahoolawee, accounted for 76% of cases for the 1990s. The last major outbreak of FBT in the state of Hawaii occurred 2002 to 2005, with Maui, Honolulu, and Kauai Counties all affected. In that four-year period, there were 168 cases (an average of 42 cases/yr), with 121, 23, and 22 cases from Maui, Honolulu, and Kauai Counties, respectively [267,268]. Concurrently, there was an apparent increase in the mouse population on the island of Maui [241]. The Hawaii Dept of Health (HDH) responded to the epidemic with both rodent control measures and public health education. The HDH applied zinc phosphide to the habitat of peri-domestic rodents, resulting in dramatic decreases in the numbers of trapped rodents on Maui and other areas. The HDH also carried out rodent trapping, environmental assessments, and rodent-proofing education in the homes of people with suspected and confirmed cases. Public awareness campaigns about FBT were mounted through physician offices, press conferences, television public service announcements, and newspaper articles [267].

On the islands of Maui, Hawaii, and Oahu there are cyclic mouse migrations every few years due to the onset of drought in rural areas, which drives the mice to seek food near human habitation [269], increasing the risk of FBT transmission. Nevertheless, the role of mice as reservoirs of FBT in the state of Hawaii is controversial. In a study conducted in 2004, 2006, and 2007, *R. typhi* and *R. felis* were detected by PCR in 2% and 25% of rat fleas, respectively, collected from house mice on Oahu; thus, *R. felis* may be a more common cause of FBT than *R. typhi* in Oahu. However, the house mice were PCR-negative for rickettsiae, suggesting that they may not be the reservoirs of FBT in Oahu, but may still serve as blood meal sources and transport hosts for rat fleas [241]. Additionally, due to the diminutive size of mice, they harbor few fleas [260]. Unlike Texas and California, in which the classic rodent-flea cycle has been mostly supplanted by the opossum-flea cycle, Hawaii has retained the former cycle. Another rodent-borne zoonosis, leptospirosis, should also be considered in the differential diagnosis of fever in Hawaii or as a co-infection in cases of FBT [261]. Possible overlapping manifestations of FBT and leptospirosis include fever, myalgias, subconjunctival hemorrhage, uveitis, renal insufficiency, and aseptic meningitis [270]. 

## 13. Conclusions 

Flea-borne typhus, due to *Rickettsia typhi* and *R. felis*, is an infection causing fever, headache, rash, and diverse organ manifestations that can result in critical illness or death. Flea-borne typhus was an emerging infectious disease, primarily in the Southern USA and California, from 1910 to 1945. The period 1930 to 1945 saw a dramatic rise in the number of reported cases; this increase was due to conditions favorable to the proliferation of rodents and their fleas during the Depression and World War II years, including: dilapidated, overcrowded housing; poor environmental sanitation; and the difficulty of importing insecticides and rodenticides during wartime. American involvement in World War II, in the short term, further perpetuated the epidemic of FBT by the increased production of food crops in the American South and by promoting crowded and unsanitary conditions in the Southern cities [4]. However, ultimately, World War II proved to be a powerful catalyst in the control of FBT by improving standards of living in the South and providing the tools for typhus control, such as synthetic insecticides and novel rodenticides. A vigorous program for the control of FBT was conducted by the US Public Health Service from 1945 to 1952. Government programs and relative economic prosperity in the South also resulted in slum clearance and improved housing. By 1953, the number of cases of FBT in the United States had dropped dramatically. Federally funded programs for rat control continued until the mid-1980s. Effective antibiotics for FBT, such as the tetracyclines, came into clinical practice in the late 1940s. The first diagnostic test for FBT, the Weil-Felix test, entered clinical practice in the 1920s, but was found to have inadequate sensitivity and specificity. The Weil-Felix test was replaced by complement fixation in the 1940s and the indirect fluorescent antibody test in the 1980s. The epidemiologic curve of FBT in the United States and the events related to the ascendency and decline of the disease from 1920 to 1987, are depicted in Figure 14. A second organism causing FBT, *R. felis*, was discovered in 1990. Flea-borne typhus persists in the United States, primarily in South and Central Texas, the Los Angeles area, and Hawaii. In the former two areas, the opossum has replaced the rat as the primary reservoir, with the cat flea as the most important vector. In Hawaii, 73% of cases occur in the county of Maui, likely because of lower rainfall than in other parts of the state. Despite the great successes against FBT in the USA in the post-World War II era, it has proved difficult to eliminate the disease because it is now associated with our companion animals, stray pets, opossums (opportunists that are able to thrive alongside humans), and the cat flea, an abundant and non-selective vector. In the new millennium, cases of FBT are increasing in Texas and California. In 2018–2019, Los Angeles County is also experiencing a resurgence of FBT, with rats as the likely reservoirs. The epidemiology of FBT in the United States over the last century exemplifies how historical events, social conditions, technology, and public health interventions influence the prevalence of a vector-borne disease.

## Figures and Tables

**Figure 1 tropicalmed-06-00002-f001:**
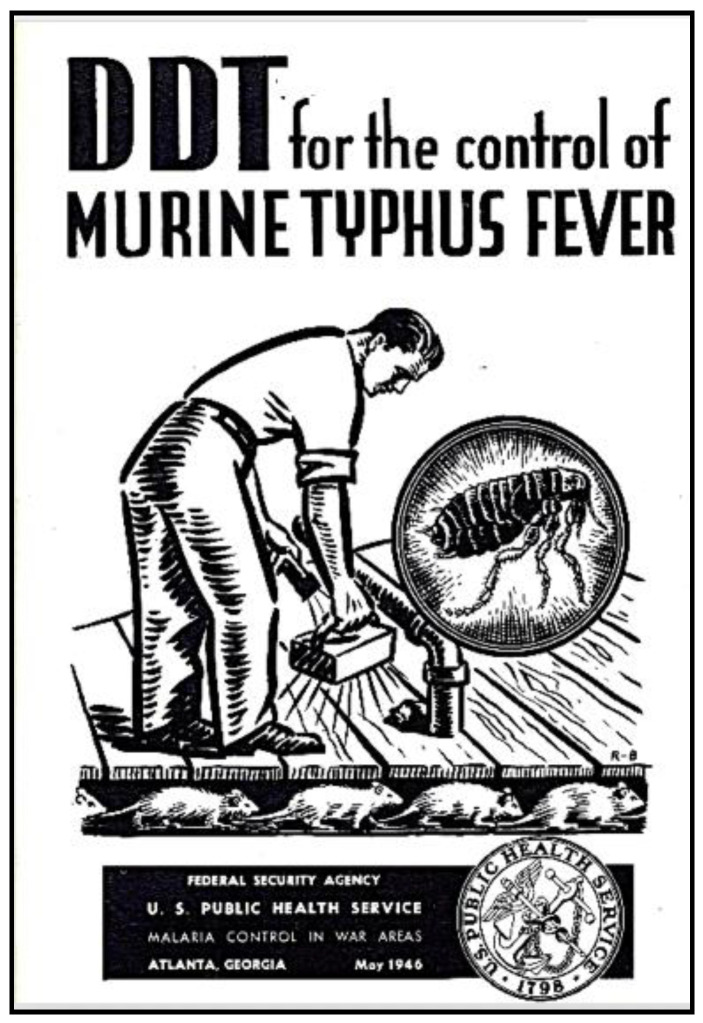
Cover of the 1946 United States Public Health Service pamphlet that first described the use of dichlorodiphenyltrichloroethane (DDT) for flea-borne typhus control [27].

**Figure 2 tropicalmed-06-00002-f002:**
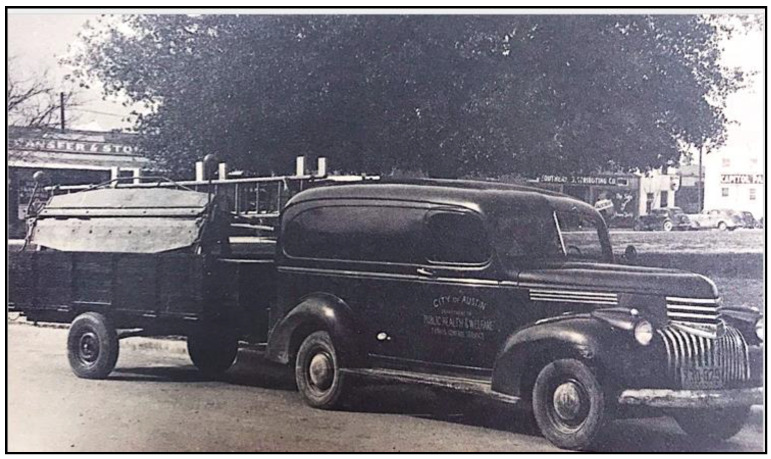
Typhus Control Truck. City of Austin, Dept of Public Health and Welfare, Typhus Control Service. Photograph published in *Texas Health Bulletin*, July 1949, with the caption “DDT dust, dispensed from spraying units such as the one pulled by the panel truck, destroyed typhus-transmitting fleas within 5–7 days after application to rat runs.”

**Figure 3 tropicalmed-06-00002-f003:**
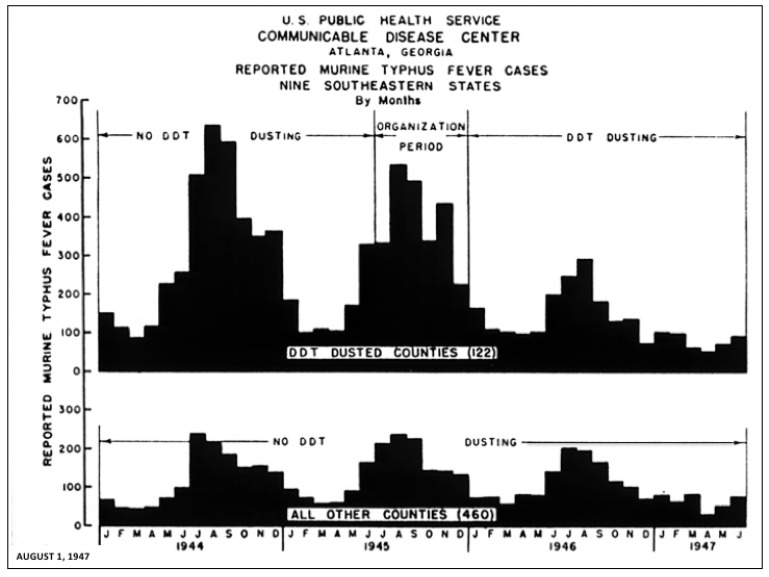
A graph showing the number of cases of flea-borne typhus in nine Southern states over time in areas treated (above) and untreated (below), showing the decrease in the treated areas, as compared to the untreated areas [28].

**Figure 4 tropicalmed-06-00002-f004:**
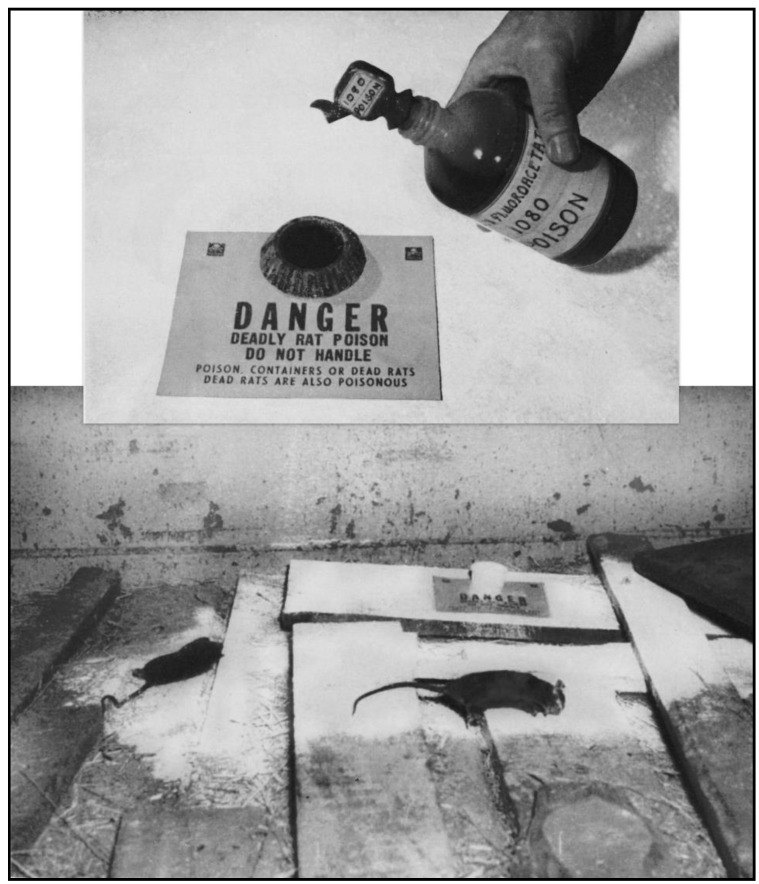
Above: Compound 1080 bait station. Below: Rats in a ship’s hold killed by 1080 (the white powder is dichlorodiphenyltrichloroethane (DDT) dust) [53].

**Figure 5 tropicalmed-06-00002-f005:**
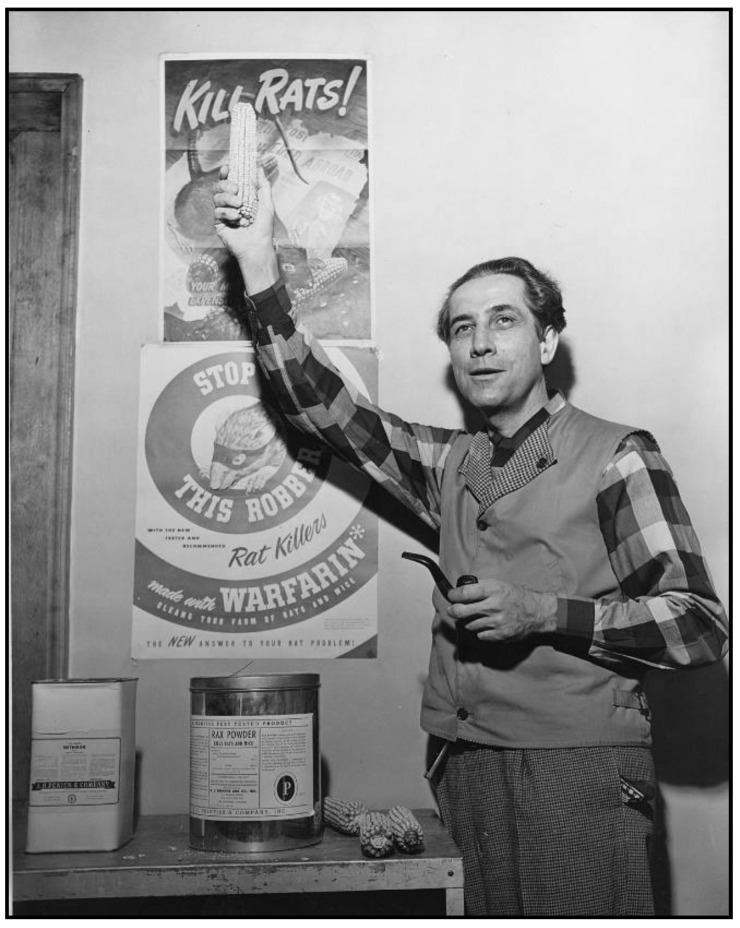
Karl Paul Link, Professor of Agricultural Chemistry at the University of Wisconsin, standing in front of a poster advertising Warfarin, a compound he and fellow researchers patented with the Wisconsin Alumni Research Foundation [81].

**Figure 6 tropicalmed-06-00002-f006:**
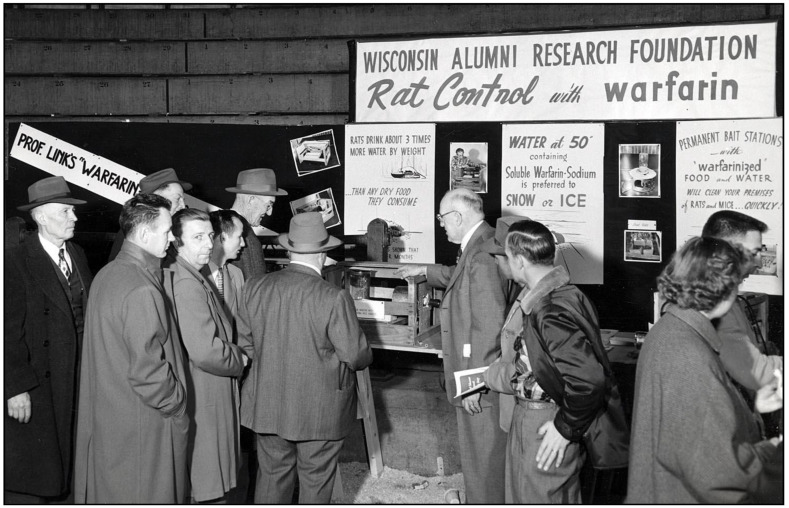
Attendees examine a poster presenting Karl Link’s work on rat extermination with warfarin during the 1954 Farm and Home Week exhibit of the University of Wisconsin College of Agriculture [82].

**Figure 7 tropicalmed-06-00002-f007:**
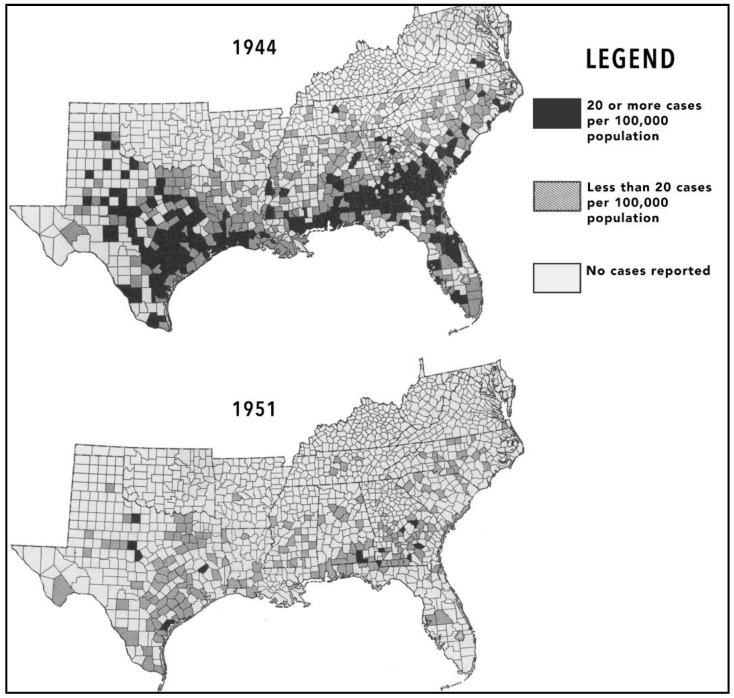
Map showing decreasing cases and geographic contraction of FBT in the United States from 1944 to 1951 [92].

**Figure 8 tropicalmed-06-00002-f008:**
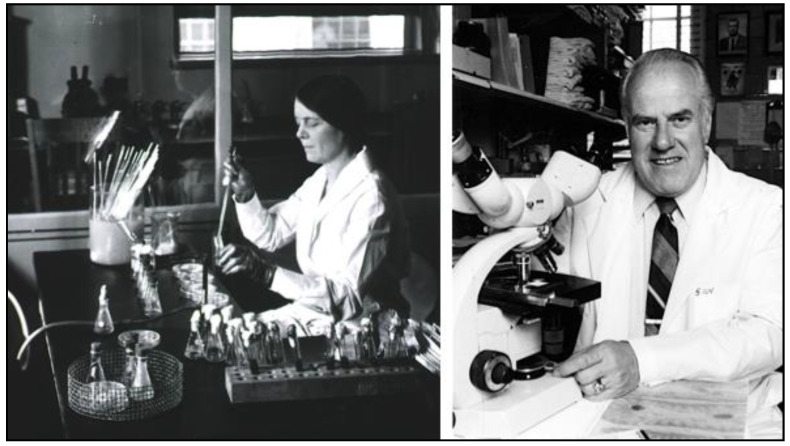
The Innovators. Left. Ida A. Bengtson PhD, the bacteriologist in the United States Public Health Service Hygienic Laboratory who developed the complement fixation serologic test for flea-borne typhus [128,129]. Right: Willy Burgdorfer PhD, director of the team at the Rocky Mountain Laboratory that developed the indirect fluorescent antibody serologic test for flea-borne typhus [134].

**Figure 9 tropicalmed-06-00002-f009:**
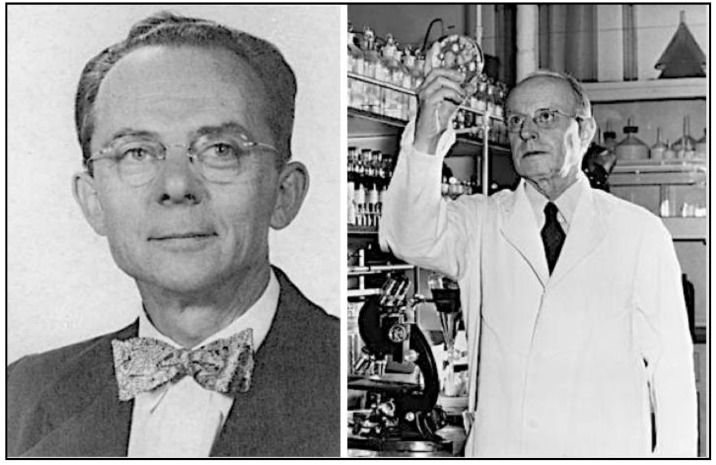
The Bioprospectors. Left. Paul R. Burkholder, the Yale University botanist who discovered chloramphenicol in a soil sample [163,164]. Right. Benjamin Duggar, the botanist at Lederle Laboratories who discovered chlortetracycline in the soil [169].

**Figure 10 tropicalmed-06-00002-f010:**
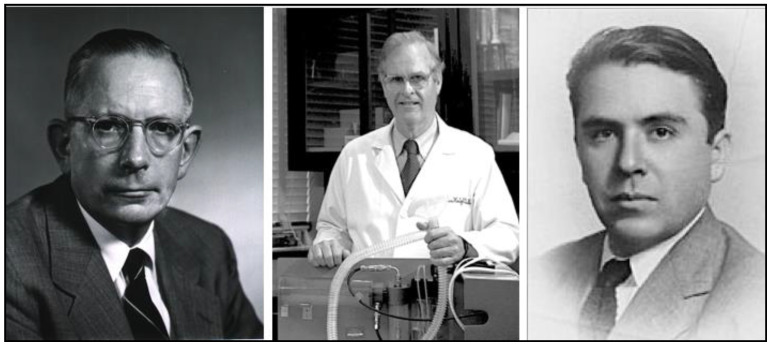
The Clinical Investigators. Left. Joseph E. Smadel MD of the Walter Reed Army Medical Center and the NIH. Smadel, Herbert Ley Jr., and Theodore E. Woodward first treated FBT patients with chloramphenicol. In 1962, Smadel received the Albert Lasker Clinical Medical Research Award “for outstanding contributions to the understanding, diagnosis, and treatment of virus and rickettsial diseases…” [174]. Center. Vernon Knight MD. Along with Francisco Ruiz-Sanchez, Amado Ruiz-Sanchez, and Walsh McDermott, Knight conducted the first trial of chlortetracycline against FBT [171]. Right. Francisco Ruiz-Sanchez MD [175].

**Figure 11 tropicalmed-06-00002-f011:**
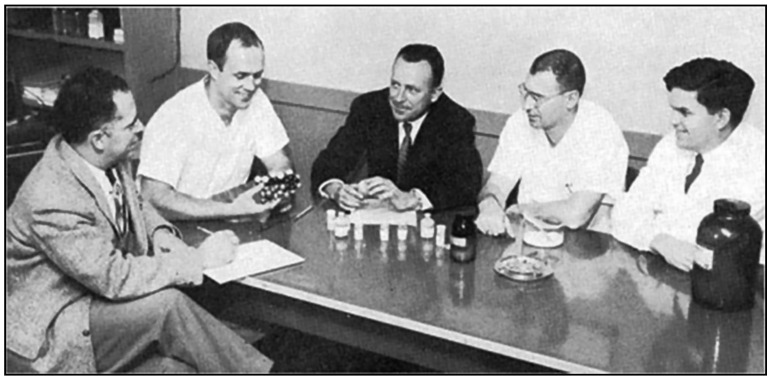
The Chemists. Pfizer Research in Brooklyn, NY. Members of the tetracycline structure determination and synthesis team (left to right): Frederick Pilgrim, Lloyd Conover, Karl Brunings, Phil Gordon, and Charles Stephens [167].

**Figure 12 tropicalmed-06-00002-f012:**
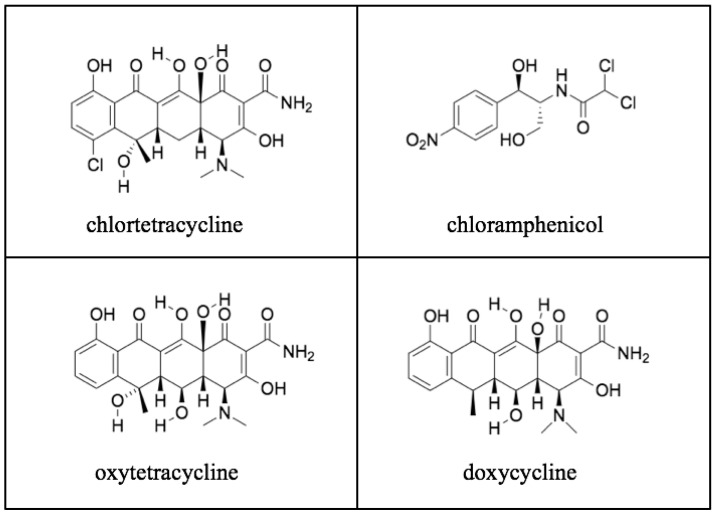
The New Antibiotics: chloramphenicol (1947), chlortetracycline (1948), oxytetracycline (1948), and doxycycline, synthesized in two steps from oxytetracycline, in 1967.

**Figure 13 tropicalmed-06-00002-f013:**
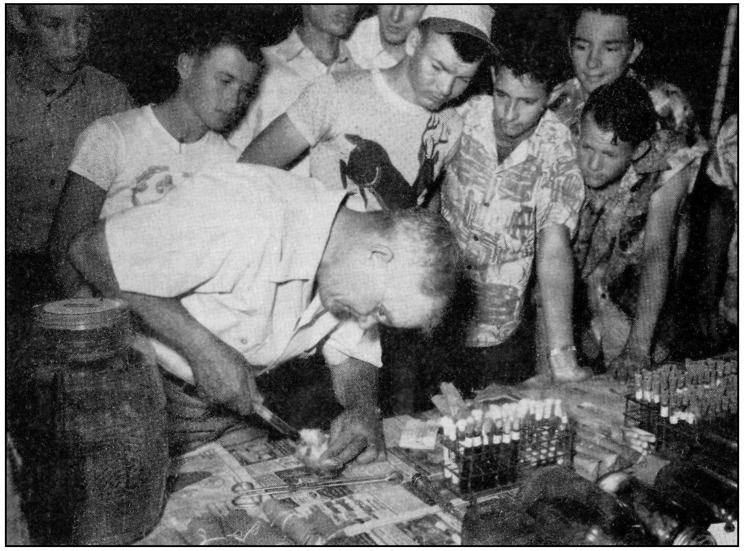
The Students and their Mentor. The original caption read: “A group of high school agricultural students watch a typhus control technician bleed rats which they trapped on their farms. The 36 boys in the Future Farmers of America chapter trapped 80 rats in three nights” [204]. The jars are filled with dead rats.

**Figure 14 tropicalmed-06-00002-f014:**
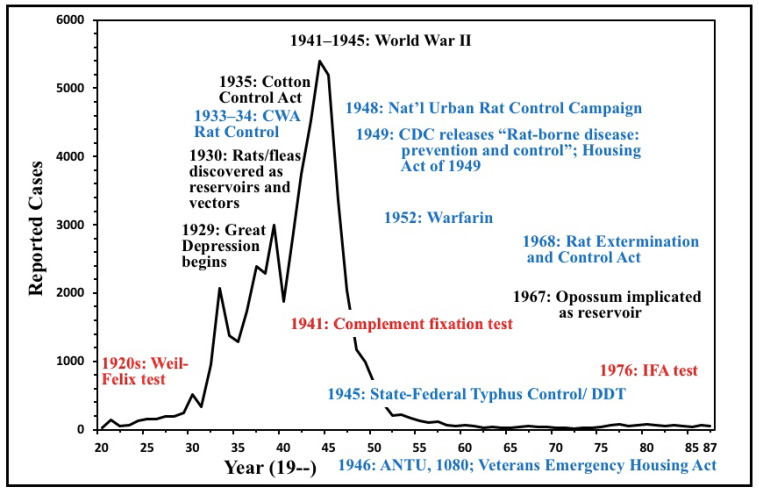
Epidemiologic curve of flea-borne typhus in the United States, 1920–1987 [2]. There are superimposed time points for: major developments in diagnostic testing (red); historical events that affected the epidemiology [4] (black); and the introduction of technologies and programs for typhus control (blue). (CWA is the Civil Works Administration).

**Table 1 tropicalmed-06-00002-t001:** Reported Cases of Flea-Borne Typhus the United States, 1920–1987 ^a,b,c^.

YEAR	Cases	YEAR	Cases	YEAR	Cases	YEAR	Cases
**1920**	31	**1940**	1878	**1960**	68	**1980**	81
**1921**	143	**1941**	2784	**1961**	46	**1981**	59
**1922**	48	**1942**	3736	**1962**	32	**1982**	58
**1923**	65	**1943**	4528	**1963**	35	**1983**	62
**1924**	130	**1944**	5401	**1964**	30	**1984**	53
**1925**	154	**1945**	5193	**1965**	28	**1985**	37
**1926**	160	**1946**	3365	**1966**	33	**1986**	67
**1927**	199	**1947**	2050	**1967**	52	**1987**	49
**1928**	196	**1948**	1171	**1968**	36		
**1929**	239	**1949**	985	**1969**	36		
**1930**	511	**1950**	685	**1970**	27		
**1931**	333	**1951**	378	**1971**	23		
**1932**	957	**1952**	205	**1972**	18		
**1933**	2070	**1953**	221	**1973**	32		
**1934**	1374	**1954**	163	**1974**	26		
**1935**	1287	**1955**	135	**1975**	44		
**1936**	1733	**1956**	98	**1976**	69		
**1937**	2394	**1957**	114	**1977**	76		
**1938**	2294	**1958**	71	**1978**	46		
**1939**	2996	**1959**	51	**1979**	69		

^a^ 1920–1967 [9]. ^b^ 1930–1987 [8]. ^c^ No national data collected after 1987.

**Table 2 tropicalmed-06-00002-t002:** Diagnostic Tests for Flea-Borne Typhus.

Test	Year Devised	Advantages	Disadvantages
Weil-Felix Test (WFt)	1915 [119]	requires minimal equipment; generally positive in the first week of infection [120]	two or more sequential sera were needed for better accuracy [122,123]; cross-reaction between rickettsial infections [120]; poor sensitivity and specificity [124]
Complement Fixation (CF)	1936 [126]; not practical until 1941 [129]	able to differentiate species of rickettsiae; CF antibodies may be present up to ≥5 years after the illness [141].	delayed positivity (second week) [120]; technically difficult [131]; lower sensitivity than IFA [133]
Indirect Immuno-fluorescence Assay (IFA)	1976 [134]	considered current gold standard; IgG sensitivity ≥83%; specificity ≥ 93% [142]; median half-life of *R. typhi* IgG was 177 days [143]	paired sera for confirmation; negative results during the first 7–14 days of infection; cross-reaction with other rickettsiae [138]; requires fluorescence microscope and reference laboratory
Latex Agglutination	1995 [137]	rapid; requires minimal equipment	less sensitive than IFA [144]
Enzyme-linked Immunosorbent Assay (ELISA)	1977 [144]	rapid; requires minimal equipment	comparable sensitive to IFA in some studies [144], but inadequate validation [138]
Polymerase Chain Reaction (PCR)	2007 [139]	potential for early diagnosis	low sensitivity when using blood samples [140,143]
Loop-Mediated Isothermal Amplification	2014 [140]	potential for rapid, point-of-care assay; does not require thermocycler	low sensitivity when using blood samples (48%) [140]

**Table 3 tropicalmed-06-00002-t003:** Cases of Flea-borne Typhus in Texas from 1943 to 2018. ^a–f.^

YEAR	Cases	Decade Total	YEAR	Cases	Decade Total	YEAR	Cases	Decade Total
**1940**	B	−	**1970**	16	321	**2000**	53	994
**1941**	B	−	**1971**	17		**2001**	29	
**1942**	1204 ^c^	8625 (1942–1949)	**1972**	13		**2002**	53	
**1943**	1452 ^d^		**1973**	28		**2003**	30	
**1944**	1740 ^d^		**1974**	12		**2004**	66	
**1945**	1844 ^e^		**1975**	30		**2005**	100	
**1946**	1147		**1976**	58		**2006**	146	
**1947**	610		**1977**	55		**2007**	169	
**1948**	344		**1978**	33		**2008**	157	
**1949**	284		**1979**	59		**2009**	191	
**1950**	222	805	**1980**	61	436	**2010**	135	3159 (2010–2018)
**1951**	164		**1981**	50		**2011**	286	
**1952**	84		**1982**	41		**2012**	263	
**1953**	82		**1983**	46		**2013**	222	
**1954**	64		**1984**	37		**2014**	308	
**1955**	52		**1985**	25		**2015**	324	
**1956**	51		**1986**	52		**2016**	364	
**1957**	38		**1987**	34		**2017**	519 ^f^	
**1958**	30		**1988**	30		**2018**	738 ^f^	
**1959**	18		**1989**	30		**2019**	^g^	−
**1960**	50	246	**1990**	36	348			
**1961**	21		**1991**	22				
**1962**	12		**1992**	18				
**1963**	21		**1993**	12				
**1964**	15		**1994**	9				
**1965**	18		**1995**	53				
**1966**	20		**1996**	41				
**1967**	37		**1997**	70				
**1968**	23		**1998**	45				
**1969**	29		**1999**	42				

^a^ Texas Dept of Health Services [203], unless otherwise indicated. ^b^ No official data. ^c^ [116]. ^d^ [21]. ^e^ [204]. ^f^ [205]. ^g^ No official data yet.

**Table 4 tropicalmed-06-00002-t004:** Cases of Flea-borne Typhus by County in the State of Hawaii, 1990–2018. ^a,b,c^.

YEAR	Hawaii	Honolulu	Kuaui	Maui	TOTAL	Decade Total	Avg No. Cases/Yr
1990	0	2	0	3	5	42	4.2
1991	0	1	0	5	6		
1992	0	0	0	3	3		
1993	0	1	0	0	1		
1994	0	0	0	5	5		
1995	0	1	0	2	3		
1996	0	1	0	4	5		
1997	0	0	0	3	3		
1998	0	0	4	5	9		
1999	0	0	0	2	2		
2000	0	2	2	1	5	235	23.5
2001	0	2	0	2	4		
2002	1	3	2	41	48		
2003	0	13	11	18	42		
2004	0	5	7	20	32		
2005	1	2	2	42	47		
2006	0	1	1	16	18		
2007	0	3	3	12	18		
2008	0	0	0	10	10		
2009	0	3	0	8	11		
2010	0	1	0	0	1	65 (2010–2018)	7.2 (2010–2018)
2011	0	1	0	13	14		
2012	0	3	0	6	9		
2013	0	0	0	4	4		
2014	0	2	0	5	7		
2015	0	3	0	5	8		
2016	0	2	1	3	6		
2017	0	3	1	4	8		
2108	1	3	0	4	8		
2019 ^d^	------	------	------	------	------		
TOTAL	3	58	34	246	341	341	11.8
% of total	0.9	17.0	10.0	72.1	100		

^a^ Hawaii Dept of Health [268]. ^b^ Kauai Co. includes the islands of Kauai and Niihau. ^c^ Maui Co. includes the islands of Maui, Lanai, Molokai, and Kahoolawee. ^d^ No official data.

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
