# Peer review of "History, Rats, Fleas, and Opossums. II. The Decline and Resurgence of Flea-Borne Typhus in the United States, 1945–2019"

_tropicalmed, 2020, doi:10.3390/tropicalmed6010002_

Round 1

Reviewer 1 Report

This manuscript reviews the history of murine typhus in the United States and is a second part to a series of the historical epidemiology of flea borne typhus (FBT).  The article is comprehensive, well written and provides a thorough historical summary of FBT in the US.  

Minor comments

  1. Line 102, "trebled" should be "tripled". 
  2. Line 175, past tense "was" rather than "is" should be used.
  3. Fig 2., legend, correct Thyphus Control Service.
  4. Line 484, change continue to continued.
  5. Line 484, change varies to varied.
  6. Line 486, delete "has"
  7. Lines 526-528, remove extra .....
  8. Line 858, font size is inconsistent

Author Response

Minor comments

  1. Line 102, "trebled" should be "tripled"; OK, done
  2. Line 175, past tense "was" rather than "is" should be used; ok, done.
  3. Fig 2., legend, correct Thyphus Control Service.
  4. Line 484, change continue to continued.  Actually, present tense for this is what was intended.
  5. Line 484, change varies to varied.  Not changed, present tense was intended.
  6. Line 486, delete "has";  not changed, an ongoing action is indicated. 
  7. Lines 526-528, remove extra .....  Ok, two sets were eliminated for improved readability.
  8. Line 858, font size is inconsistent.  All font sizes were made consistent throughout the manuscript.

Reviewer 2 Report

The article is very well written and gives an extremely detailed and comprehensive overview of Flea-borne-typhus in the era 1940-2019. All content is well fuelled with adequate references. Content-wise, there are no remarks.

It’s an editorial decision how many pages the manuscript should have. From a readers perspective, the article can be perceived as very long, and maybe the number of pages number could be decreased. If a shorter version is desired, I would suggest to shorten paragraph 3 (The Advent of New Rodenticides and Integrated Rodent Control) and the 3 final paragraphs that focus on Texas, LA and Hawai.

It would be nice if Figure 14 could be adapted, so that it is up-to-date until 2019.

2 small typing errors:

Line 71: reserviors

Line 1226: that able to

Author Response

It’s an editorial decision how many pages the manuscript should have. From a readers perspective, the article can be perceived as very long, and maybe the number of pages number could be decreased. If a shorter version is desired, I would suggest to shorten paragraph 3 (The Advent of New Rodenticides and Integrated Rodent Control) and the 3 final paragraphs that focus on Texas, LA and Hawaii.

I went through the manuscript and eliminated excessive verbiage, without sacrificing the informational content and compromising the writing style, of which all of the reviewers gave a favorable assessment.

It would be nice if Figure 14 could be adapted, so that it is up-to-date until 2019.

No official national data was collected after 1987, which is why the graph stops at 1987.   

2 small typing errors:

Line 71: reservoirs. Corrected.

Line 1226: that able to.  Corrected to: that are able to

Reviewer 3 Report

This manuscript is a fascinating account of the historical epidemiology of flea-borne typhus in the USA, very well written and illustrated. I really enjoyed read it and I am sure that it will attract many readers.

I have no major comments, but only a very minor one. Could the author add one sentence in the Summary to present the main aim of his manuscript and it relation with his previous one? I think it will help the reading.

Author Response

I have no major comments, but only a very minor one. Could the author add one sentence in the Summary to present the main aim of his manuscript and it relation with his previous one? I think it will help the reading.

The following sentence was added to the Summary:

This is the second part of a two-part series describing the rise, decline, and resurgence of FBT in the United States over the last century.  These studies illustrate the influence of historical events, social conditions, technology, and public health interventions on the prevalence of a vector-borne disease.